# The distinct translational landscapes of gram-negative *Salmonella* and gram-positive *Listeria*

Owain J. Bryant[1,2,3], Filip Lastovka [1,3], Jessica Powell[1] & Betty Y. -W. Chung [1] ✉

Translational control in pathogenic bacteria is fundamental to gene expression and affects virulence and other infection phenotypes. We used an enhanced ribosome profiling protocol coupled with parallel transcriptomics to capture accurately the global translatome of two evolutionarily distant pathogenic bacteria—the Gram-negative bacterium *Salmonella* and the Gram-positive bacterium *Listeria*. We find that the two bacteria use different mechanisms to translationally regulate protein synthesis. In *Salmonella*, in addition to the expected correlation between translational efficiency and *cis*-regulatory features such as Shine–Dalgarno (SD) strength and RNA secondary structure around the initiation codon, our data reveal an effect of the 2nd and 3rd codons, where the presence of tandem lysine codons (AAA-AAA) enhances translation in both *Salmonella* and *E. coli*. Strikingly, none of these features are seen in efficiently translated *Listeria* transcripts. Instead, approximately 20% of efficiently translated *Listeria* genes exhibit 70 S footprints seven nt upstream of the authentic start codon, suggesting that these genes may be subject to a novel translational initiation mechanism. Our results show that SD strength is not a direct hallmark of translational efficiency in all bacteria. Instead, *Listeria* has evolved additional mechanisms to control gene expression level that are distinct from those utilised by *Salmonella* and *E. coli*.

Global transcriptome quantification techniques (e.g., RNASeq) are powerful methods to study the regulation of physiology and pathogenesis of *Salmonella*[1–6]. These studies have revealed that, despite the close linkage between prokaryotic transcription and translation, transcript levels often do not correlate with protein abundances[7–13].

Studies on translational control in bacteria have predominantly been based on the Gram-negative bacterium *E. coli* as a model organism and have shown that a combination of Shine–Dalgarno (SD) strength and optimal codon usage amongst other factors contribute to efficient translation[14–16]. Despite these important studies, translational control in other Gram-negative bacteria differs greatly, for example some phyla such as *Aquificota* and *Bacteroidota* naturally lack SD sequences[17–19]. Gram-positive bacteria are even more evolutionarily

distant, and their translational control is poorly understood. Both *Listeria* and *Salmonella* have undergone divergent evolutionary paths. They have different nutrient requirements, and variations in nutrient limitations have been known to influence translation control in *E. coli*[20–22]. They also exhibit differences in motility and adaptability ranges[23,24]. Hence, some of the differences in translation regulation in these key pathogenic species might be related to the distinction between motile and non-motile bacteria rather than Gram-positive or Gram-negative categorization.

Given that a significant number of infection-related deaths in humans are associated with bacterial pathogens (both resistant and susceptible to antimicrobials)[25], it is crucial that we have a better understanding of bacterial translational regulation. We therefore

[1]Department of Pathology, University of Cambridge, Tennis Court Road, Cambridge CB2 1QP, UK. [2]Present address: Centre for Structural Biology, National Cancer Institute, 21702 Frederick, MD, USA. [3]These authors contributed equally: Owain J. Bryant, Filip Lastovka. ✉e-mail: bcy23@cam.ac.uk

investigated two representative bacterial pathogens—the Gram-negative *Salmonella enterica* serovar Typhimurium (*S.* Typhimurium) and the Gram-positive *Listeria monocytogenes* (*L. monocytogenes*), to understand the global translation landscape and its relationship with transcriptional control, particularly for the virulence machinery.

Non-typhoidal serovars of *Salmonella* are the leading cause of food-borne gastroenteritis; they infect millions and cause c. 230,000 fatalities each year[26]. *S.* Typhimurium is the most common non-typhoidal *Salmonella* strain isolated from patients around the world and is used in mouse models to study bacterial pathogenesis and host-microbe interactions by Gram-negative pathogens[27,28]. In addition to infecting humans, *S.* Typhimurium is an important pathogen in live-stock including chickens, pigs and cattle[29,30]. It utilises a multitude of virulence factors to reach and invade host cells, and to support its intracellular survival[31–34].

Most *Salmonella* virulence factors are encoded within horizon-tally acquired genomic regions known as *Salmonella* pathogenicity islands (SPIs)[35–39]. SPI-1 encodes a type III secretion system (T3SS), also known as an injectisome, which is responsible for the trigger-based invasion mechanism utilised also by many other Gram-negative bac-teria. The SPI-1 T3SS penetrates host cells and secretes effector pro-teins through the needle to facilitate invasion of intestinal cells[31,37,38,40,41]. A further four pathogenicity islands (SPI-2, 3, 4, and 5) encode virulence factors required for infection and survival within host cells[38,39,42]. In addition to SPI-encoded virulence factors, some virulence factors are encoded outside of SPIs. Key examples are genes which encode additional effector proteins (e.g., SopA and SopE), which promote bacterial entry into host cells, regulate bacterial survival within host cells and control inflammatory responses[31,43,44]. Another key virulence factor is the bacterial flagellum, a large macromolecular rotary motor that enables cell motility, including to sites of infection[45–47]. A major component of the flagellum is the protein, fla-gellin, an important antigen which triggers a range of immune responses in host cells[48–52].

The Gram-positive bacterium *Listeria monocytogenes*, the leading cause of listeriosis, is one of the most virulent foodborne pathogens, with a high rate of death associated with infection. Like *Salmonella*, *Listeria* encodes a range of virulence factors to promote entry and survival within host cells. Upon invasion of host cells, these bacteria replicate within the phagosome and produce listerolysin O (LLO), which lyses the phagolysosomal membrane, allowing the bacteria to escape into the cytoplasm, where they proliferate. They use cell-surface virulence factors to promote host actin polymerisation and thus mediate actin-based motility within the cytoplasm and into neighbouring cells, allowing dissemination within host tissues. Inter-estingly, LLO levels are translationally regulated to control virulence, highlighting the role of this control mechanism on bacterial pathogenesis[53].

To investigate translational control in *Salmonella* and *Listeria*, we utilised ribosome profiling (RiboSeq) which involves deep sequencing of ribosome protected mRNA fragments to directly capture protein synthesis in a natural setting. Initially developed in yeast, RiboSeq is a highly sensitive global method that reveals the translatome at the time of harvest[54]. The technique determines the positions of ribosomes by exploiting the protection of a discrete fragment of mRNA (~30 nucleotides) from nuclease digestion conferred by a translating ribosome[8,54]. Deep sequencing of these ribosome-protected fragments (RPFs) generates a high-resolution view of the location and abundance of translating ribosomes on different mRNA species. The combination of RNASeq and RiboSeq datasets provides a global picture of: (1) mRNA abundance; (2) total protein synthesis (i.e., total number of ribosome-protected fragments on all mRNAs per gene—a direct measurement of the total amount of proteins being synthesised at the time of harvest); and (3) translation efficiency, a measurement of how well each mRNA is being translated (i.e., number of ribosome-protected fragments per

mRNA; calculated by dividing the number of phased RPFs mapped to the coding region normalised to the total number of phased RPFs in all coding regions and the number of RNA-Seq reads mapped to the coding region normalised to the total number of RNA-Seq reads mapped to all coding regions; see methods for information about quantification). However, obtaining high quality RiboSeq datasets from bacteria has been problematic due to numerous technical diffi-culties and, so far, there are limited wild-type bacterial ribosome profiling data of sufficient resolution to reflect the triplet periodicity of decoding, which is instrumental to accurately identify short ORFs and non-canonical control mechanisms[55]. Here, we optimised RiboSeq for both model intracellular pathogens *Salmonella enterica* and *Listeria monocytogenes*. Our data display precise triplet phasing and, for the first time, allow us to determined their global translatome, permitting an accurate global characterisation of bacterial translational control.

## Results
### Measurement of the steady-state translatomes of *Salmonella* and *Listeria* with high-definition RiboSeq
To dissect the regulatory layers of virulence and pathogenicity of *Salmonella* and *Listeria*, we generated highly phased RiboSeq and parallel RNASeq data (Figs. 1A, S1), allowing us to accurately uncouple RNA abundance from translation efficiency of cellular components directly relevant to virulence. The total translation (protein synthesis) of a given gene is dependent on both its mRNA abundance and the efficiency with which it is translated. As we were interested in com-ponents immediately relevant to infection and virulence, cells were harvested at $OD_{600}$ of 1, when peak production of the SPI-1 transcrip-tional master regulator, HilA, occurs in *Salmonella*, and in the expo-nential growth phase for *Listeria*[56,57]. These conditions allowed us to capture translation within *Salmonella* and *Listeria* that are 'primed' for infection[58].

In addition, we also addressed a major obstacle in bacterial ribo-some profiling: the removal of rRNA species, which account for a sig-nificant proportion of the total reads (>95%), restricting analysis to highly abundant transcripts unless immense resources are dedicated to sequencing[59]. We therefore utilised a combination of rRNA sub-traction using subtraction kits based on oligonucleotides anti-sense to bacterial rRNA, and duplex-specific nuclease (DSN)-based depletion treatment, to substantially reduce the proportion of rRNA reads within libraries, thereby substantially enriching libraries for reads corre-sponding to ribosome-protected fragments of mRNAs[59,60] (Fig. 1B, Table S1). While the combination of anti-sense rRNA-based subtraction and one round of DSN treatment was effective at depleting reads corresponding to non-coding RNAs (ncRNA) including rRNA from *Salmonella* RiboSeq libraries, *Listeria* RiboSeq libraries required two further rounds of DSN treatment to sufficiently enrich for reads cor-responding to RPFs (Figs. 1B, S2E).

Upon obtaining substantial sequencing depth, ~74% (3450/4682) of *Salmonella* genes and ~77% (2155/2800) of *Listeria* genes passed our filtering criteria for downstream analysis (see methods and Table S1). Quality control analysis was performed with particular attention to commonly known artefacts in bacterial RiboSeq, which could lead to over-interpretation of data[61]. First, artefacts in bacterial RiboSeq stu-dies can arise due to the choice of nuclease used to generate RPFs[10,62]. RNase I is typically used to generate RPFs from eukaryotes, whereas its use in *E. coli* has been less successful, reportedly due to its inhibition by *E. coli* ribosomes[63]. For this reason, bacterial RiboSeq studies tend to use S7 micrococcal nuclease (MNase). However, S7 MNase exhibits significant sequence specificity, resulting in RiboSeq data that contains high levels of noise and a lack of triplet phasing[61]. We titrated S7 MNase and RNase I to determine the optimal nuclease concentration to gen-erate RPFs (Fig. S2). Metagene translatome analysis revealed accu-mulation of reads corresponding to translation initiation, regardless of nuclease treatment (Figs. 1A, S2B). Similar to previous reports, we

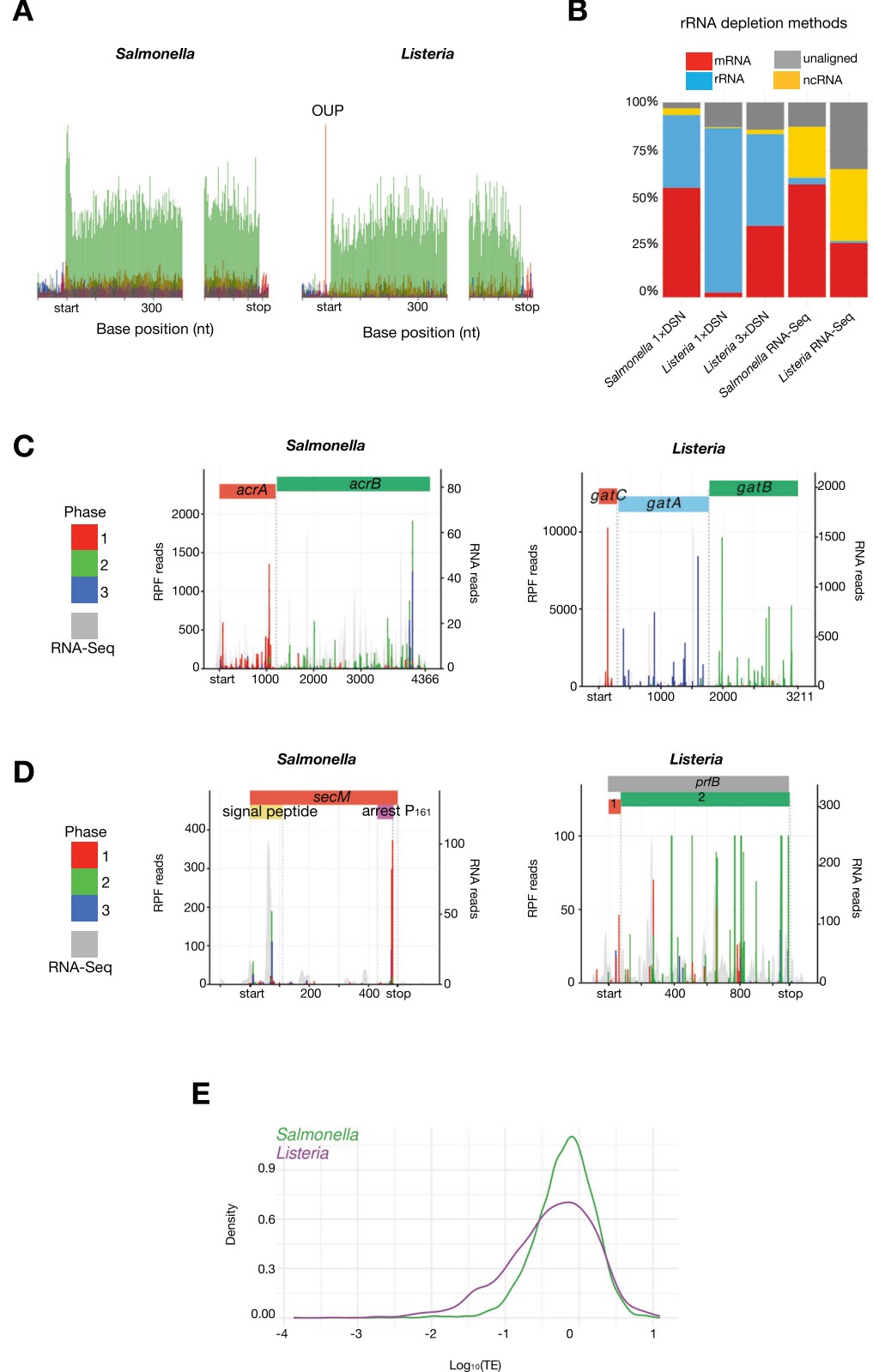

found that treatment with S7 MNase resulted in data that do not reflect triplet phasing[9,10,61,62] (Fig. S2B). In contrary to previous work in *E. coli*, treatment with RNase I, however, resulted in RPFs with a distinct size distribution that are highly phased, indicating that the enzyme can be used to generate *Salmonella* and *Listeria* RiboSeq data with single nucleotide resolution, visible at the individual gene level (Figs. 1A, D, 2A, C, S3–5). Further, despite, a correlation of protein synthesis levels

of genes in libraries generated with S7 MNase and RNase I, it was apparent that S7 MNase-treated RiboSeq datasets tended to over-estimate protein synthesis, especially for genes which are poorly translated (Fig. S2D).

We then focused on in-frame reads, i.e., reads that genuinely reflects ribosome occupation, for all downstream analysis, complemented by parallel RNASeq to tease apart RNA abundance (i.e., total

**Fig. 1 | Tandem-RNASeq and RiboSeq of *Salmonella* and *Listeria* cells.**
**A** Metagene translatome plots generated by aligning all coding sequences using the start and stop codons as anchors. Reads from the *Salmonella* SL1344 RiboSeq library (left) or *Listeria* 10403 S RiboSeq library (right) that map to codon positions 1, 2, and 3 are coloured in red, green and blue, respectively. The plot was produced with the R software package riboSeqR[60]. **B** Library composition of RiboSeq libraries from *Salmonella* (SL1344) or *Listeria* (10403 S) generated with RNase I that were subjected to oligonucleotide-based rRNA subtraction, followed by one or three rounds of DSN treatment. The composition of RNASeq libraries post

oligonucleotide-based rRNA subtraction is on the right. **C** Visualisation of translation of polycistronic transcripts: the *Salmonella acrA-acrB* operon, the *Listeria gatC-gatA-gatB* operon. **D** the *Salmonella secM* gene, which utilises programmed ribosomal pausing; and the *Listeria prfB* gene, which utilises +1 programmed ribosome frameshifting. Red, green and blue bars indicate RiboSeq reads mapping to phases 1, 2, and 3, respectively, relative to the first nucleotide of the transcript. Grey shaded peaks show parallel RNASeq data. **E** Distribution of translation efficiency for all *Salmonella* (green) and *Listeria* (purple) genes.

RNASeq) from protein synthesis (i.e., total RiboSeq). We processed the data in a similar manner to our previous studies[60,64,65], followed by correlation of protein synthesis with RNA abundance. As expected, protein synthesis positively correlated with RNA abundance. Importantly, we also observed that a large number of mRNAs were synthesised but inefficiently translated (Figs. S2C, S2G, S2I). This supports the notion that RNASeq alone does not accurately reflect global protein levels in bacteria despite close coupling of transcription and translation[48].

We were also able to identify distinct features of *Salmonella* and *Listeria* translation. First, while the quality of *Salmonella* and *Listeria* RiboSeq libraries are comparable (Figs. 1A, S3), a subset of genes in *Listeria* had a large number of ribosomal footprints where the expected P-site maps to a position seven nucleotides upstream of the start codon; we termed these 'out-of-frame upstream peaks' (OUP, Fig. 1A, right). These OUP reads are absent in all *Salmonella* RiboSeq libraries. Sequence enrichment analysis revealed that *Listeria* genes containing the OUP on average do not have a significantly stronger SD sequence than *Listeria* genes without the OUP (Fig. 4E). Secondly, we observed that RPFs of highly abundant read sizes (24, 25, 27, 28, and 29 nucleotides) in *Salmonella* are more phased than other read sizes (Fig. S3), similar to previous reports of plant and algae (ribosome profiles[60,64,66,67]. This observation was consistent with a library generated with a different *Salmonella* strain as well as *Listeria* on a separate occasion (Fig. S3)[60], suggesting that the accessibility to nucleases of the mRNA exit channel of *Salmonella* and *Listeria* ribosomes is similar to that of *Arabidopsis* and *Chlamydomonas*, where they are less protected by ribosomal proteins than for mouse and human ribosomes. In contrast to the RiboSeq libraries, reads in our RNA-Seq datasets have the expected features, including broader read-size distributions, are not phased, and are equally enriched in the untranslated regions and coding regions of mRNAs[54,59,60,64,65,68] (Figs. S2B, S2F, S2H S3A, S3B).

The phased *Salmonella* and *Listeria* RiboSeq data also enabled direct visualisation of intrinsic features of bacterial translation, as illustrated by the following examples. First, the reading frames of individual genes were clearly visible, as exemplified by the *acrAB* operon in *Salmonella* (Fig. 1C, left) and the *gatCAB* operon in *Listeria* (Fig. 1C, right). Secondly, we could detect ribosome pausing events such as following the ribosome arrest motif in *Salmonella secM* (Fig. 1D, left). Thirdly, we could detect changes in ribosome density at the known sites of programmed ribosomal frameshifting within the *Salmonella prfB* and *dnaX* genes and the *Listeria prfB* gene[69–73] (Fig. S5, Fig. 1D, right). Thus the translation of previously unannotated ORFs in *Salmonella* and *Listeria* is detected with high confidence (Table S2, Fig. S6). Finally, we revealed that global distribution of translation efficiency of *Salmonella* is more compressed than *Listeria* (Fig. 1E).

### Stoichiometric production through coordinated translation of protein complexes associated with virulence

We next assessed whether protein synthesis levels correlate with stoichiometry of components in multiprotein complexes. Components within multiprotein complexes are typically thought to be produced at levels which match their stoichiometry, as the excess production of one or more components over another could result in

incorrect assembly, misfolding or aggregation, besides resource wastage[74,75]. Li et al.[9] showed that ~55% of components in multiprotein complexes in *E. coli* are produced at levels indistinguishable from their stoichiometry. In *Salmonella* too, overall protein copy number positively correlates with translation efficiency, supported by the stronger correlation with protein synthesis than RNA abundance, suggesting that correct stoichiometry for a significant subset of proteins is determined translationally (Fig. S7, Table S3).

We observed many of structural-component-related operons where protein copy number is directly controlled during translation (i.e., via translation efficiency). We therefore further investigated how these operons—many of which are associated with pathogenicity—are translationally controlled, starting with ATPase complexes. We first looked at the $F_1F_o$ ATPase complex, which in *Salmonella* is encoded by the *atp* operon. It consists of nine genes, eight of which encode subunits that assemble to form the $F_1F_o$ ATPase complex (Fig. 2A). The $F_1F_o$ ATPase is expressed from a single polycistronic transcript and should require individual genes within the operon to exhibit different translational efficiencies to reflect the stoichiometry of the complex[9]. *atpE* was removed from these analyses due to an overwhelming accumulation in read counts due to bias in both the RNAseq and RiboSeq libraries, (Fig. S8A, B). Similar to *E. coli*, we also observed a positive correlation between stoichiometry and translation efficiency in both *Salmonella* and *Listeria*, indicating that utilisation of translational control as a mechanism for proportional protein synthesis and stoichiometric assembly of the $F_1F_o$ ATPase complex is a common feature of both Gram-positive and Gram-negative bacteria[9] (Fig. 2B). We also found that stoichiometric production of components of the multi-drug efflux components AcrAB and the iron-sulfur assembly components SufBCD are primarily regulated at the level of translation (Figs. S9A–C, S10). In contrast, stoichiometric production of components of the virulence T3SS ATPase complex is regulated by both transcript abundance and translation efficiency (Fig. S9D).

To better understand expression strategies utilised for the production of virulence factors, we assessed whether the stoichiometry of components within multiprotein complexes associated with virulence correlates with RNA abundance (transcription or RNA turnover) or translation efficiency and assessed whether the correlation is supported by the functions of individual components (e.g., whether T3SS effectors are expressed in excess of the injectisome structure through which they are secreted). We first examined the *Salmonella* SPI-1 locus, which, unlike the $F_1F_o$ ATPase, is composed of multiple operons, and we focused specifically on the structural components that assemble to form the SPI-1 injectisome type III secretion system (vT3SS)—a large multiprotein complex of known stoichiometry, ranging from one subunit copy to over 100 copies, that is essential for active host invasion (Fig. 2C, D). In contrast to $F_1F_o$ ATPase, stoichiometry of the majority of the T3SS structural components correlates well with transcript abundance, likely due to the expression derived from eight polycistronic transcripts, but more strongly with total protein synthesis, thus positive correlation with TE, suggesting that stoichiometry of SPI-1 structural components is regulated both transcriptionally and translationally (Fig. 2C, D).

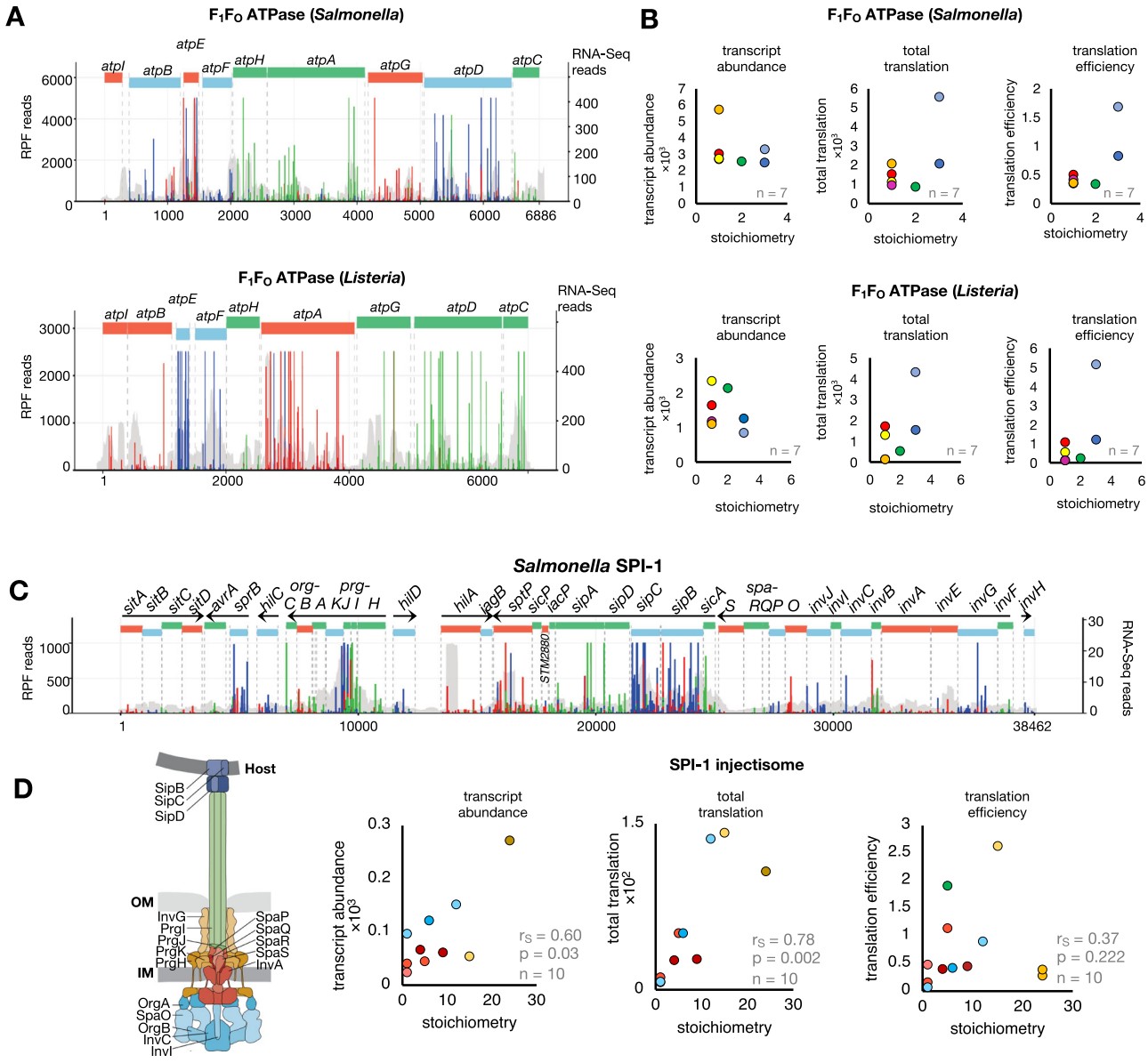

**Fig. 2 | Proportional synthesis of protein complexes associated with virulence.**
**A** Visualisation of translation of *Salmonella* and *Listeria* atp operon, which contains the genes encoding components of the $F_1F_O$-ATPase complex (top and bottom, respectively). Red, green and blue bars indicate RiboSeq reads mapping to frames 1, 2, and 3, respectively. Grey shaded peaks show parallel RNASeq data. RiboSeq axis adjusted to enable visualisation of less translated ORFs. **B** Relationship between stoichiometry of $F_1F_O$ ATPase complex subunits and the corresponding transcript abundance, protein synthesis or translation efficiency. The data point corresponding to the atpE gene is an outlier due to significant bias and was excluded from correlation analyses. **C** Visualisation of translation of the *Salmonella* SPI-1

regulon which is composed of 8 transcripts (annotated with black arrows where arrow heads indicate the direction of transcription) and codes for virulence genes, including structural components of the SPI-1 injectisome. Red, green and blue bars indicate RiboSeq reads mapping to frames 1, 2, and 3, respectively. Grey shaded peaks show parallel RNASeq data. **D** Relationship between stoichiometry of SPI-1 vT3SS structural genes and the corresponding transcript abundance, protein synthesis or translation efficiency (Right). Schematic representation of the SPI-1 vT3SS highlighting structural proteins (Left). Schematic modified from Wagner et al. [103].

## Regulation of proportional protein synthesis in *Salmonella* and *Listeria*

As our ribosome profiling revealed that the stoichiometry of many subunits within protein complexes in *Salmonella* and *Listeria* is translationally controlled, we reasoned that translation efficiency could be controlled by a combination of one or more factors. One is the strength of the SD sequence, which provides an indication of initiation efficiency and is calculated based on modelled thermodynamics of ribosome–mRNA interaction, which considers both the mRNA secondary structure and the strength of 16 S rRNA and SD sequence hybridisation[76]. The second factor is codon usage, which may affect

regulation during elongation, and a third is secondary structure throughout the mRNA. To investigate these factors, we first determined the codon adaptation index (CAI) and SD strength of all translated *Salmonella* and *Listeria* genes and compared either CAI or SD strength score with transcript abundance, protein synthesis, or translation efficiency (Figs. 3A, B, S12, 13). Consistent with efficient initiation being the limiting factor for protein synthesis, the correlation of SD strength with translation efficiency is stronger in both *Salmonella* and *Listeria* than with transcript abundance or protein synthesis. This was particularly marked in the case of *Salmonella*. In contrast, correlation between CAI and translation efficiency is stronger in *Listeria* compared

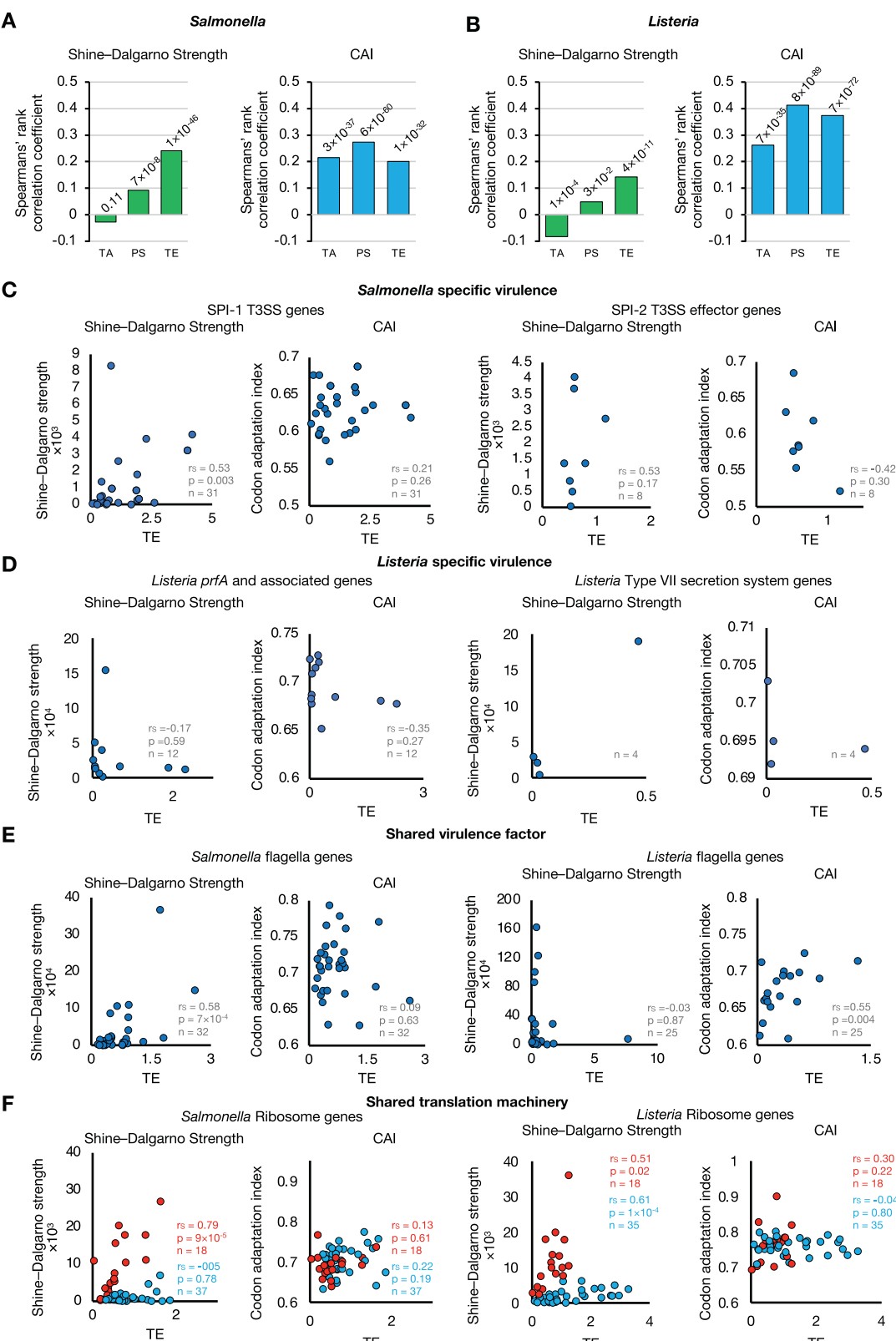

**Fig. 3 | Regulation of proportional protein synthesis in *Salmonella* and *Listeria*.** **A**, **B** Bar charts showing Spearman's rank correlation coefficients for the relationship between SD strength (left) or codon adaptation index (CAI, right) with transcript abundance (TA), protein synthesis (PS) or translation efficiency (TE) of all *Salmonella* and *Listeria* genes, respectively. The approximate *p* values have been calculated and are displayed above each bar. **C** SD strength (left) or codon adaptation index (CAI, right) plotted as a function of translation efficiency for

*Salmonella* specific virulence factors: *Salmonella* SPI-1 T3SS structural genes (left) or *Salmonella* SPI-2 effector genes (right). **D** *Listeria* specific virulence factors: *Listeria prfA* and associated genes (left) or *Listeria* type VII secretion system genes (right). **E** Shared virulence factor: *Salmonella* (left) and *Listeria* (right) flagella genes. **F** Shared translation machinery: *Salmonella* (left) and *Listeria* (right) ribosomal protein genes. Genes groups with distinct correlation between translation efficiency and SD strength are coloured in red or blue.

to *Salmonella*. As expected, little to no correlation was seen between transcript abundance and SD strength for either *Salmonella* or *Listeria* (Fig. 3A, B).

We next restricted the analysis to just *Salmonella* and *Listeria* virulence genes that encode components of multiprotein complexes (Figs. 3C–F, S8–S11). For the SPI T3SS genes of *Salmonella*, we observed a strong positive correlation between SD strength and translation efficiency of both SPI-1 T3SS structural and effector genes and SPI-2 T3SS effector genes while correlation with CAI was poor, suggesting that regulation for virulence machinery is largely determined through regulating translational initiation efficiency (Fig. 3C). As for virulence genes of *Listeria*, we observed no correlation between translation efficiency and either SD strength or CAI for *Listeria prfA* and associated genes, which was expected as these genes are not part of a single virulence apparatus, although we also did not observe correlations for *Listeria* Type VII secretion system genes (Fig. 3D).

We next focused on multiprotein complexes that exist in both *Salmonella* and *Listeria*, starting with $F_1F_o$ ATPase, *sufB–D* and flagella (Fig. 3E–F, Figs. S9–10, S14, 15). While we generally observed positive correlation between SD strength and translation efficiency in *Salmonella*, we observed no correlation for either SD strength or CAI for any of these complexes in *Listeria*, with the exception of flagella genes where translation efficiency positively correlates with CAI (Fig. 3E). Finally, we examined one of the most evolutionarily conserved multiprotein complexes, the ribosome (Fig. 3F). In *Salmonella*, we identified two groups of ribosomal proteins where translation efficiency either strongly correlates with SD strength (red), or not at all (blue) by stratifying the data points based on whether they lay above or below $y = m*x + c$ in the Shine–Dalgarno vs translation efficiency plots (Fig. 3F, Fig. S15). In contrast, while there are also two groups of ribosomal proteins in *Listeria*, both groups display a positive correlation between SD strength and translation efficiency with one group being more sensitive to the SD strength (in blue, Figs. 3F and S16C). Overall, these results indicate that regulation of translation initiation in *Samonella* is more directly influenced by SD strength compared to *Listeria*.

### *Cis*-regulatory elements that facilitate efficient translation in bacteria

Because the SD strength depends on both the mRNA sequence that interacts with the 16 S rRNA and the mRNA secondary structure strength, we separately investigated the relationship of both factors with translation efficiency. First, we hypothesised that transcripts that are inefficiently translated have a preference to form stronger RNA secondary structure, which may impede accessibility of the SD sequence to the 16 S rRNA[77–80]. To test this, we grouped genes into four groups, comprising the top 100 most or least abundant transcripts or efficiently translated genes (i.e., 2.9% and 4.6% of translated genes in *Salmonella* and *Listeria*, respectively). We then predicted the average minimum free energy (MFE) of RNA secondary structure throughout the transcript for each group of genes within a 30-nt-wide sliding window[64] (Fig. 4A, B). These analyses revealed that *Salmonella* mRNAs that are efficiently translated tend to be less structured around the start codon compared to transcripts that are not efficiently translated, as expected (Figs. 4A, S18). This is consistent with previous reports that reduced RNA secondary structure at start codons influences translation efficiency[81–84]. Interestingly, we also noted that weakly structured 3′ UTR is a distinct feature for *Salmonella* mRNAs that are low in abundance or inefficiently translated (Figs. 4A, S18). Compared with *Salmonella*, *Listeria* transcripts are generally less structured throughout the 5′ UTR and CDS, with mfe ≈ −2 until 3′ UTR. In addition, highly abundant *Listeria* transcripts tend to be more structured in the coding region and 3′ UTR (Figs. 4B, S19). However, we only saw a marginally weaker structure between the start codon and ~35 nt upstream, a marginally stronger 3′ UTR structure of efficiently translated *Listeria* mRNAs and comparable structure strength in the coding sequence of

efficiently translated *Listeria* genes (Fig. 4B), despite the difference between the 100 most and least efficiently translated *Listeria* transcripts being much greater than *Salmonella* (Fig. 1E).

### *Listeria* genes containing out-of-frame upstream peaks (OUPs) are more efficiently translated

While there is a lack of global preference for weak 5′ UTR secondary structure in efficiently translated *Listeria* transcripts, we observed a highly unusual feature in a subset of such genes in that they contain a high accumulation of out-of-frame 22–24 nt footprints where the P-site is inferred to lie seven nucleotides upstream of the start codon (Fig. 4J). These out-of-frame upstream peaks (OUPs) were observed in ~16% of all translated genes and were only observed in *Listeria* RiboSeq libraries and not in *Salmonella* RiboSeq libraries (Figs. 1A, 4C, D, S3B, S4A). Nucleotide sequence logos revealed that *Listeria* genes containing an OUP contain on average a marginally stronger SD sequence and more often initiate with an AUG codon compared to non-OUP *Listeria* genes (Figs. 4E, S20A, and S20G). We also found that the distance between the SD sequence and the start codon does not differ between OUP and non-OUP genes, suggesting that the spacer is not a determinant of the OUP (Fig. S20C). In addition, the site of the OUP-footprint is not significantly less structured than in non-OUP genes (Fig. S20F). Importantly, comparison between OUP and non-OUP genes showed that mRNAs of OUP genes are more abundant and more efficiently translated than mRNAs of non-OUP genes (Figs. 4F, S4B). Moreover, OUP genes generally encode proteins involved in regulation of translation, transcription, response to stimulation or localisation/membrane transporters (Figs. 4G, S20D–E). Despite comprising just 16% of genes (355/2155), OUP genes were responsible for 40% of all protein production (Fig. 4G).

To gain further insight into OUP-mediated translation, we next investigated the read size distribution of different classes of ribosome protected footprints, namely initiation (where inferred P-site aligns to the start codon) and elongation footprints in both *Salmonella* and *Listeria* and OUP in *Listeria* (Fig. 4H–J). As expected, the size distribution of *Salmonella* initiation and elongation footprints peaks at 28–30 nt, and 24 nt, respectively (Fig. 4H). The larger *Salmonella* initiation footprint is likely due to the presence of initiation factors 1 and 2 during initiation and/or alternative ribosome conformation present when the SD sequence interacts with the anti-SD sequence within the 30 S subunit, together with positioning of the start codon at the P-site. A previous report identified longer ribosome footprints at internal Shine–Dalgarno sequences for elongating ribosomes, and proposed that ribosomes may pause at internal SD sequences whilst the ribosomes decode a few subsequent codons[85]. Intriguingly, in *Listeria*, the size distribution of OUP footprints is ~22–24 nt, similar to elongation footprints for both OUP and non-OUP genes, as well as a small proportion of non-OUP initiation footprints, while initiation footprints for OUP genes are significantly larger at ~29–32 nt compared to initiation footprint for non-OUP genes (20, 23 and 27 nt) (Fig. 4I).

### The sequence context surrounding the start codon has a greater impact on *Salmonella* translation than for *Listeria*

To further investigate additional *cis*-regulatory elements that may contribute to translation efficiency, whether during initiation or at the early stages of elongation, we investigated sequences surrounding the start codons, both in terms of nucleotide and amino acid identity. Conservation of sequences upstream of and flanking start codons of the most efficiently translated genes in *Salmonella* revealed that the SD sequence has a preference for an adenosine followed by two tandem guanosine nucleotides at positions −10 to −8 compared to inefficiently translated genes (Fig. 5A). In contrast, efficiently translated *Listeria* genes have a preference for adenosine at position −3 and −4, a feature similar to the eukaryotic Kozak consensus sequence[86,87], but only minor differences in SD sequence compared to genes that are

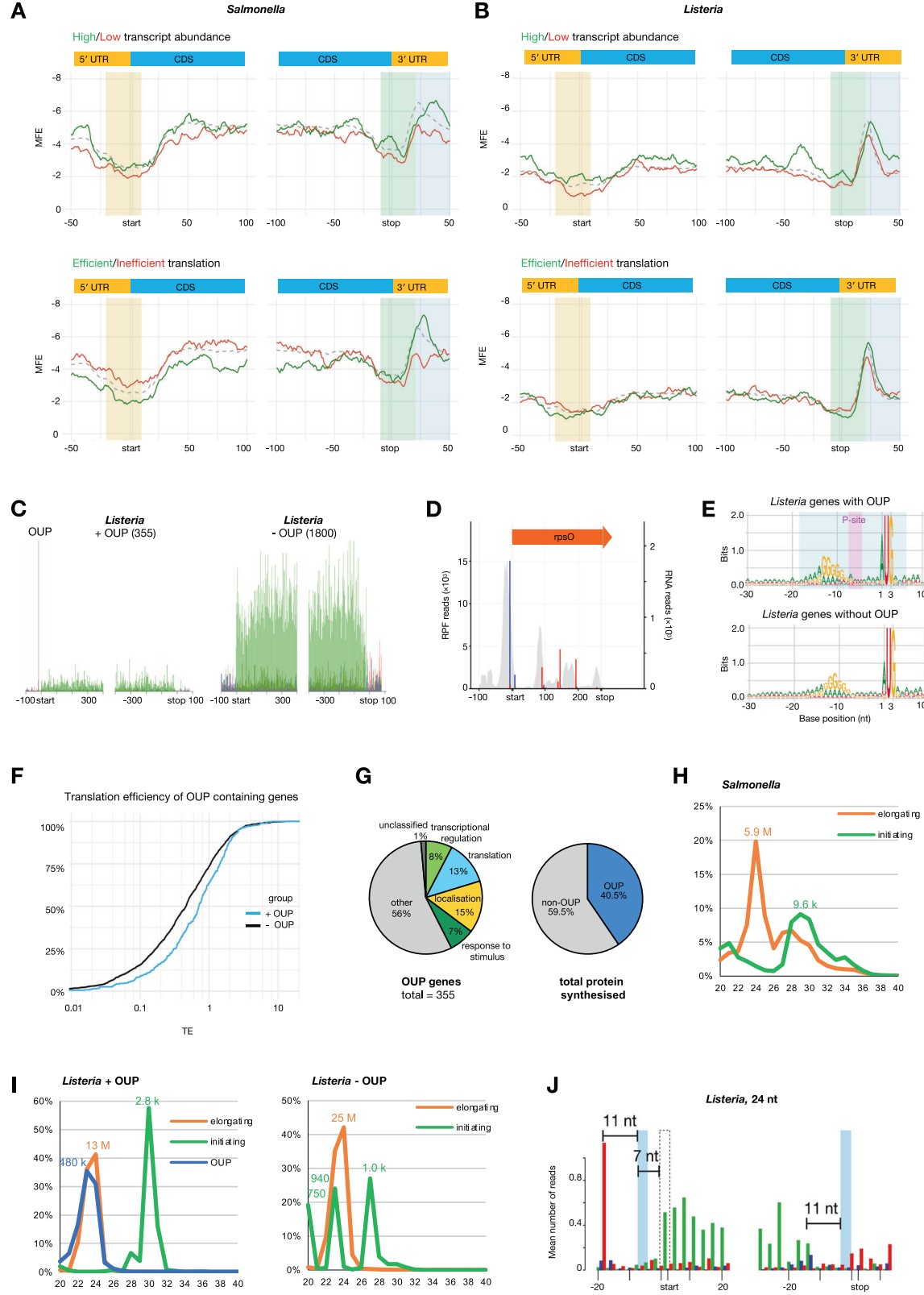

inefficiently translated (Fig. 5B). This complements our analysis above where SD strength is a major determinant of translation efficiency in *Salmonella*. In *Listeria*, however, the less clear difference in nucleotide preference in the SD sequence as well as local secondary structure between efficiently and inefficiently translated genes complements our observation of poor correlation between SD strength, translation efficiency and stoichiometry (Figs. 3B, D, S13). Furthermore, our

*Salmonella* data revealed that SD sequences of inefficiently translated transcripts tend to be further away from the start codon compared to those of efficiently translated transcripts (Fig. S20B), consistent with previous reports in *E. coli*, and supporting the notion of spacer length being an important modulator of initiation efficiency[15,76,88]. However, we did not observe differences in spacer length for *Listeria*, except perhaps for a small proportion of OUP-genes where the SD sequence is

**Fig. 4 | RNA secondary structure, OUPs and their link with translation efficiency of genes in Salmonella and Listeria. A, B** Sliding-window average MFE of predicted secondary structure at each nucleotide of sequences surrounding the start and stop codons for the 100 most abundant (green) or least abundant (red) transcripts (top) or the 100 most efficiently translated (green) or least efficiently translated (red) transcripts (bottom) in *Salmonella* (**A**) or *Listeria* (**B**). Secondary structure in the highlighted areas is further compared in Fig. S18 and 19. **C** Metagene translatome plots generated by aligning all open reading frames of *Listeria* with the abundant out-of-frame upstream peak (OUP) of at least ten reads (left) or without (right) using the start and stop codons as anchors. Reads that map to codon positions 1, 2, and 3 are coloured in red, green and blue, respectively. **D** Visualisation of translation of a *Listeria* OUP-gene *rpsO*. Red, green and blue bars indicate RiboSeq reads mapping to frames 1, 2, and 3, respectively. Grey shaded peaks show parallel RNASeq data. **E** Nucleotide sequence logos of all *Listeria* genes with an upstream out-of-frame peak (OUP) of at least ten reads (top) and for all *Listeria* genes without OUP (bottom). The OUP footprint (24 nt) is highlighted in blue whilst the P-site is highlighted in magenta. **F** The cumulative distribution for the translation efficiency of OUP genes (blue) and non-OUP genes (black). **G** Pie chart of gene ontology groups of OUP genes (left). The 'response to stimulus' group excludes genes that also belong to 'transcriptional regulation', 'translation' and 'localisation'. Overlaps are illustrated in Fig. S20E. Pie chart on the right reflects the proportion of all protein synthesised from OUP and non-OUP genes. **H** Read size distribution from *Salmonella* RiboSeq libraries of initiating (green) or elongating (orange) ribosome footprints. **I** Read size distribution from *Listeria* RiboSeq libraries of OUP genes (left) or non-OUP genes (right). Initiating, elongating, and OUP RPF lengths are represented in green, orange or blue, respectively. **J** Detailed metagene translatome plot of *Listeria* 10403 S generated by aligning all coding sequences using start and stop codons as anchors. The P-site position calculated from OUPs and the P-site position of terminating ribosomes are illustrated in light blue, and their distances to the 5′-most peaks are specified. The start codon position is illustrated as a dashed box and it is located 7 nt downstream of the calculated OUP P-site position. Compare with figure S4.

1 nt further away (S20B–C). These observations highlight the distinct differences in the regulation of translation initiation between the two evolutionarily divergent bacteria.

Unexpectedly, we also detected an enrichment for adenosine at the 4th, 5th, 7th and 8th nucleotides in the coding region of efficiently translated genes in *Salmonella*, which would correspond to an enrichment of either asparagine (AAU, AAC) or lysine (AAA, AAG) residues at the 2nd and 3rd amino acid positions (Fig. 5A). A similar enrichment for adenosine nucleotides was observed in the coding region immediately following the start codon in *Listeria*, although there is similar enrichment also for inefficiently translated genes (Fig. 5B). Previous studies using reporter constructs revealed that lysine residues encoded by codons 3 and 5 enhance reporter activity[89]. Subsequent work showed that genes that natively encode the nucleotide combinations that enhanced protein expression in the reporter constructs exhibited only a marginal increased translation efficiency[90]. Further studies have studied the role of codons at early stages of elongation using reporters with variable outcomes[91–95]. Inspection of the amino acid sequences encoded by genes that are efficiently translated revealed an enrichment for lysine residues over asparagine residues at both sites (Figs. 5C, D, S21). As lysine residues contain positively charged side chains, we assessed whether there is also a bias towards coding for positively charged arginine residues at the 2nd and 3rd codon positions in efficiently translated genes. To do this we stratified all *Salmonella* and *Listeria* genes based on whether they code for two tandem arginine, lysine or asparagine residues at the 2nd and 3rd codon positions. We found that in *Salmonella*, genes coding for two tandem lysine residues at the 2nd and 3rd positions were translated more efficiently than average and, interestingly, genes coding for two tandem arginine residues at the 2nd and 3rd positions were translated less efficiently than average, despite reported similar lysine and arginine tRNA abundances and both amino acids containing positively charged side chains[96] (Fig. 5C). The translation efficiency of genes encoding two tandem asparagine codons was comparable to average. In contrast, we did not observe a meaningfully higher translation efficiency of genes that encode two tandem lysine codons at the 2nd and 3rd codon positions in *Listeria* (Fig. 5D, right).

To test whether the presence of lysine, arginine or asparagine codons at the 2nd and 3rd codon positions influences translation efficiency we utilised a luciferase (LuxAB) translational reporter assay where transcription is controlled by a tetracycline-inducible promoter, in both *Salmonella* and the Gram-negative *E. coli*. Strikingly we found that the presence of two lysine codons (AAA-AAA) at the 5′ end of the *luxA* coding region significantly increases reporter luminescence, which was not observed for two arginine (CGU-CGU) nor for two asparagine (AAU-AAU) codons (Fig. 5E). The luminescence increase for the di-lysine-codon-containing reporter was apparent shortly after transcriptional induction, and transcript levels remained similar between all constructs throughout the time course (Fig. S22).

To further investigate the effect on translation efficiency of di-lysine (AAA-AAA), di-asparagine (AAU-AAU) or di-arginine (CGU-CGU) at the 2nd and 3rd codon positions, we fused to *luxA* the first 10 codons of the wild-type *Salmonella* gene, *ynfB*, (which natively begins with AUG-AAU-AAU) or its variants in which the tandem AAU-AAU are replaced with AAA-AAA or CGU-CGU. Luciferase reporters containing the AAA-AAA replacement gave significantly higher luminescence than reporters containing the arginine replacement or the wildtype AAU-AAU sequence (Fig. 5F, left). Similarly, we fused to *luxA* the first 10 codons of the wild-type *Salmonella* gene, *yceI*, (which natively begins with AUG-AAA-AAA) or its variants in which the tandem AAA-AAA are replaced with AAU-AAU or CGU-CGU (Fig. 5F, right). Reporter constructs where the wild-type AAA-AAA sequence is replaced with tandem AAU or CGU codons gave lower luminescence than reporters containing the wild type sequence. Overall, these results are in agreement with our RiboSeq data where presence of two lysine codons AAA-AAA at the 5′ end of ORFs correlates with efficient translation and show that this feature is functional in both *Salmonella* and *E. coli*.

## Discussion

Through accurate determination of translation efficiency of all expressed mRNAs in *Salmonella* and *Listeria*, we have identified multiple regulatory mechanisms that contribute towards efficient translation and virulence factor production. Moreover, we have shown that stoichiometric control of components of multiprotein complexes that drive virulence is hardwired within the messenger RNAs to directly control differential translation in *Salmonella* but not in *Listeria*.

We confirmed that a strong SD sequence as well as weak RNA secondary structure around the start codon are instrumental for efficient translation in *Salmonella*, presumably to enable efficient 30 S subunit joining to the messenger RNA. In addition, we discovered that presence of AAA at the 2nd and 3rd codons of ORFs drives efficient translation in both *Salmonella* and *E. coli*; perhaps this sequence facilitates efficient transition of the 70 S ribosome from initiation to elongation (Fig. 6). However, regulation of translation initiation differs substantially in *Listeria*. Despite the greater variance of *Listeria* translational efficiency compared to *Salmonella*, the average SD sequences between efficiently and inefficiently translated transcripts are almost indistinguishable, and there is also little difference in average secondary structure throughout the mRNA. However, this is likely a consequence of low GC% of the *Listeria* genome (37.8% compared to 52.2% in *Salmonella*), and therefore does not need to select against structure around the start codon to achieve the same lack of structure to promote efficient translation as in *Salmonella*.

Perhaps the most striking feature of efficiently translated *Listeria* genes are the OUPs, potentially indicating a novel mechanism of

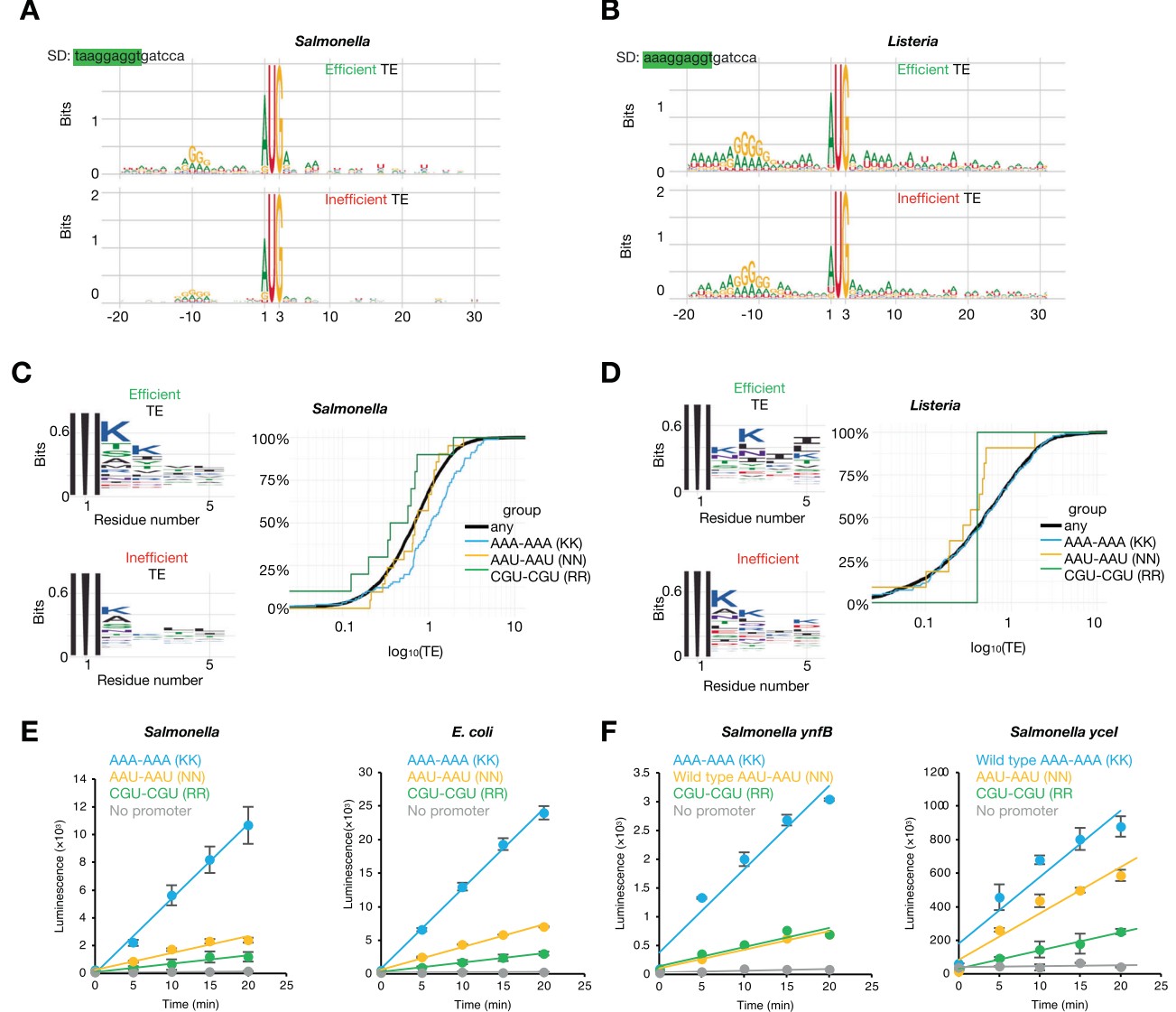

**Fig. 5 | Sequence features regulating translation efficiency of genes in *Salmonella* and *Listeria*. A**, **B** Sequence logos of sequences surrounding the start codon of the top 100 most efficiently translated genes (top) or least efficiently translated genes (bottom) in *Salmonella* (**A**) and *Listeria* (**B**), respectively. The SD sequence based on the sequence of the *Salmonella* or *Listeria* 16 S rRNA sequence is highlighted in green (top left). **C**, **D** Sequence logos of N-terminal amino acid residues of the top 100 most efficiently translated genes (top left) or least efficiently translated genes (bottom left) in *Salmonella* (**C**) and *Listeria* (**D**), respectively. The translation efficiency cumulative distribution of genes containing AAA-AAA, AAU-AAU or CGU-CGU codons following the start codon is on the right. **E** Induction time courses of LuxAB luciferase expression measured by luminescence. Constructs containing AAA-AAA (KK, blue), AAU-AAU (NN, orange) or CGU-CGU (RR, green) or a construct lacking a promoter (grey) were expressed in *Salmonella* (SL1344) or *E. coli* (MC4000) ($n = 3$). **F** Induction time courses of *luxAB* luciferase expression measured by luminescence where *luxA* is fused with the first 30 nt of *ynfB* (which natively begins with AAU-AAU after the start codon) or the first 30 nt of *yceI* (which natively begins with AAA-AAA, after the start codon) to LuxA, or their mutants (as indicated) ($n = 3$).

bacterial initiation. Canonical bacterial initiation begins with the 70 S ribosome assembly with initiation factors 1 and 2, with the start codon positioned in the P-site, results in larger initiation footprint due to presence of initiation factors, as observed in *Salmonella* (Fig. S4A). It is possible that OUP footprints derive from a 30 S subunit associated with initiation factors; however this is unlikely as pre-initiation 30 S is not thought to sufficiently protect mRNA from nuclease digestion. Another possibility is that in *Listeria*, for OUP-mediated initiation 70 S formation occurs further upstream in the absence of initiation factors 1 and 2, resulting in a similar footprint size as elongating ribosomes, followed by progression of 7 nt and acquisition of initiation factors and fMet-tRNA to initiate at the start codon, with a larger footprint size due to incorporation of initiation factors 1 and 2 (Fig. 6). A third possibility is that the ribosomes generating an OUP footprint have the start codon in the P-site, but this would necessitate structural rearrangements to

account for the observed footprint size (22–24 nt). Additional factors and/or an unusual ribosome conformation could lead to the protection of additional seven nucleotides 5′ of the P-site, and the conformation and/or empty A-site might make the mRNA 3′ end accessible to RNase I cleavage closely downstream of the P-site AUG. Future work will investigate which of these hypotheses, if any, is correct.

Translational regulation is a critical process that governs the rate and efficiency of protein synthesis in all living cells. Our study has highlighted striking differences in translational regulation between two evolutionarily divergent pathogenic bacteria. The intrinsic molecular differences in translational regulation revealed by our analysis will provide new strategies for specific targeting of antibiotic-resistant bacteria, most importantly with respect to Gram-positive pathogens. Additionally, characterisation of these differences can lead to a better understanding of the basic

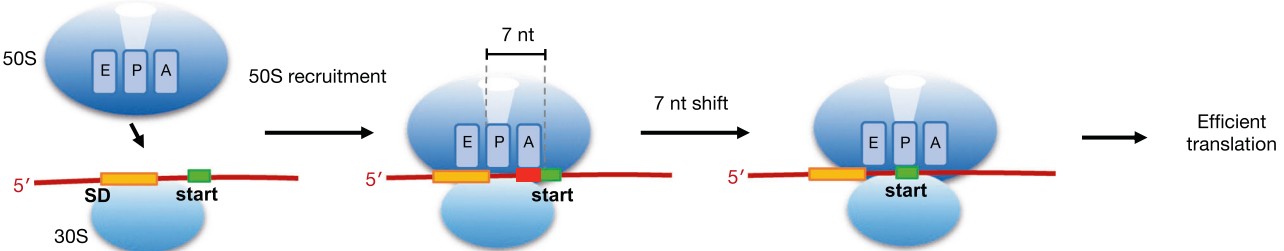

**Fig. 6 | Models of *Salmonella* and *Listeria* translation initiation.** Summary model showing *cis*-elements utilised for efficient translation in *Salmonella* (top) and *Listeria* (middle and bottom). In *Salmonella*, translation initiation is directly modulated by weak RNA secondary structure surrounding the start codon as well as interaction between the SD and anti-SD sequence. In addition, decoding of di-lysine codons AAA-AAA soon after initiation significantly enhances translation elongation. In contrast, in *Listeria*, there is no obvious preference for weak RNA secondary structure nor SD sequence nor di-lysine codons after the start codon for efficient translation. Instead, *Listeria* possibly utilises two distinct translation initiation mechanisms, a canonical mechanism where the assembly of the ribosome occurs with the start codon at the P-site while a second distinct mechanism utilised by significant number of efficiently translated genes involves assembly of the ribosome 7 nt upstream of the start codon prior to migration of 7 nt until the P-site encounters the start codon for translational initiation (bottom).

molecular biology of these organisms, which can inform broader research in the field of microbiology.

## Methods
### Bacterial growth conditions and cell harvesting for ribosome profiling

*Salmonella* strains were streaked onto an LB agar plate and incubated overnight at 37 °C. A single colony was used to inoculate LB and incubated overnight at 37 °C with shaking (180 RPM). The overnight culture was diluted 1:100 in LB (50 ml) and grown in a 250 ml flask at 37 °C with shaking (180 RPM). Once the culture reached $OD_{600}$ 1.0, 500 ml of chloramphenicol (150 mg/ml in 100% ethanol) was added to the culture for 1 minute. The culture was then poured through a funnel containing frozen LB containing chloramphenicol (1500 mg/ml) such that the culture was rapidly cooled whilst passing through the funnel

into a new 250 ml flask beneath that was kept on ice. This resulted in rapid cooling of the culture without dilution of the chloramphenicol. The rapidly cooled culture was immediately pelleted by centrifugation (6000 × *g*, 1 min, 4 °C). Cell pellets were resuspended in pre-chilled profiling buffer (1 ml, Table S4) and snap frozen with liquid nitrogen prior to −80 °C storage for later use.

*Listeria monocytogenes* 10403 S strain was streaked onto a BHI agar plate and incubated overnight at 37 °C. A single colony was used to inoculate BHI broth and incubated overnight at 37 °C with shaking (200 RPM). The overnight culture was diluted 1:10 in BHI (50 ml) and grown in a 250 ml flask at 37 °C with shaking (200 RPM). Once the culture reached $OD_{600}$ 1.0, 750 ml of chloramphenicol (100 mg/ml in 100% ethanol) was added to the culture for 1 minute. The culture was then poured through a funnel containing frozen BHI containing chloramphenicol (1500 mg/ml) such that the culture was rapidly

cooled whilst passing through the funnel into a new 250 ml flask beneath that was kept on ice. This resulted in rapid cooling of the culture without dilution of the chloramphenicol. The rapidly cooled culture was immediately pelleted by centrifugation ($6000 \times g$, 1 min, 4 °C). Cell pellets were resuspended in pre-chilled profiling buffer (1 ml, Table S4) and snap frozen with liquid nitrogen prior to −80 °C storage for later use.

## Preparation of cell extracts
Frozen cells were pulverised to a fine powder in liquid nitrogen and immediately clarified by centrifugation once thawed ($13,000 \times g$, 2 min, 4 °C). The supernatant was transferred to a new tube and the $A_{254\,nm}$ was adjusted to 10 with profiling buffer. Aliquots (100 ml) of lysate were snap frozen in liquid nitrogen and stored at −80 °C.

## Generation of RiboSeq libraries
Cell extracts (100 ml) were digested with RNase I or S7 MNase for 30 minutes at 28 °C with shaking (400 RPM). To stop the nuclease digestion, 10 ml of SUPERase·In RNase inhibitor (ThermoFisher Scientific) was added before placing the sample on ice. To purify monosomes, digested extracts were spun through pre-equilibrated S400 columns followed by TRIzol extraction. The precipitated and washed RNA pellet was dissolved in 10 mM Tris-HCl pH 7.5 prior to rRNA removal with Gram-negative or Gram-positive bacteria Ribocop rRNA depletion kit (Lexogen) for *Salmonella* or *Listeria*, respectively, or with Ribo-Zero rRNA Removal Kit (Bacteria; Illumina) followed by size selection for ribosome protected fragments and library generation. Duplex-specific nuclease treatment was applied to the dsDNA library to further deplete ribosomal RNA[60,97].

## Generation of parallel RNASeq libraries
Total RNA from an aliquot (100 ml) of cell extract extracted with TRIzol. The precipitated and washed RNA was dissolved in 10 mM Tris-HCl pH 7.5 prior to rRNA removal with Ribo-Zero rRNA Removal Kit (Bacteria; Illumina), followed by alkaline hydrolysis with fragmentation buffer (Table S4) at 95 °C for 15 minutes. Fragments in a specific size range were selected and libraries were generated in parallel with RiboSeq[60,97].

## Luciferase reporter assays
*Salmonella* cells carrying a modified pWKS30 vector encoding the *luxA* fusion and *luxB* under the control of a Tet-inducible promoter were grown in LB containing 100 mg/ml carbenicillin to $A_{600\,nm}$ 1.0 and the cell density of cultures normalised based on $A_{600nm}$. Anhydrotetracycline (ATC) was added to the cultures to a final concentration of 50 ng/ml. Aliquots (100 ml) of culture were collected at time points after induction with ATC and placed directly into a 96-well plate containing 20 ml of chloramphenicol (4 mg/ml) to inhibit further protein synthesis. To measure the luminescence of cell cultures, 30 ml of 5× luminescence solution (0.11% dodecanal, 0.2% Triton X-100) was added to each 96-well plate and mixed by pipetting. Samples were incubated at room temperature for 2 minutes before measuring luminescence in a SpectraMax M5 (Molecular Devices) using SpectraMax Pro v.5.

## RNA isolation and qPCR
Cells from 0.3 ml of *Salmonella* culture were suspended in 300 µL of phenol–chloroform pH 8. The aqueous phase was extracted and processed in the same way using phenol–chloroform pH 4.5. The RNA was co-precipitated with GlycoBlue (Thermo Fisher Scientific) and dissolved in 10 mM Tris pH 7.5. To remove any remaining genomic DNA, the solution was treated with 3 U of RNase-free DNase I (Thermo Fisher Scientific). RNA was re-purified by phenol–chloroform pH 4.5 extraction and its integrity was verified by agarose gel electrophoresis. To generate cDNA by reverse transcription, 1 µg of RNA was mixed with 250 pmol of random hexadeoxyribonucleotides (Promega) and

10 nmol of dNTPs, and annealed by heating to 70 °C for 5 min followed by cooling to 4 °C. First strand synthesis reactions were prepared by adding 40 U of RNase OUT (Invitrogen), 200 U of M-MLV reverse transcriptase and M-MLV reverse transcription buffer (Promega). Reverse-transcriptase-free samples were prepared as control groups. The reactions were performed at 37 °C for 1 hour and inactivated at 70 °C for 15 min. qPCR reaction mixes were prepared in a 384-well plate with primers aligning to either *luxA* (sequences: CCTTATGTTGCTGCCGCACACC and TTGTCGAACCGGATGGGCAGTC; efficiency: 1.765) or *lppA* (SL1344_1311; sequences: GTACTAAACTGG-TACTGGGCG and GCTGATCGATTTTAGCGTTGC; efficiency: 1.929). Each 10 µL sample consisted of 4 µL of reverse transcription product diluted to 1/10 in nuclease-free water, 500 nM of forward and reverse qPCR primer, 5 µL of iTaq Universal SYBR Green Supermix (Bio-Rad). A qPCR was set up in a QuantStudio 5 Real-Time PCR system. Relative *luxA* transcript levels were calculated using the Pfaffl method[98] with MKK at 0 minutes post induction as a reference sample and *lppA* as a reference housekeeping gene.

## Bioinformatics
Raw fastq files were processed as previously described[60,64,65,67] and mapped to reference genomes LT2 (accession: NC_003197.2 and NC_003277.2), SL1344 (accession: FQ312003.1, HE654725.1, HE654726.1, and HE654724.1) or 10403 S (accession: NC_017544.1). RNA and RPF reads mapping to CDSs were counted, filtered: >50 RPF reads at minimum of 10 unique positions prior further processing. Expression analysis normalised by CDS length and library size (RPKM, reads per kilobase per million mapped reads). Read length ranges used for gene expression analyses were: 23–30 nt for *Salmonella* and 23–25 nt for *Listeria* (fig. S3B). Translational efficiency was calculated by normalized RPF/RNA mapped to the coding region. Metatranslatome plots were generated with *riboSeqR*[60,64,65]. Single gene maps were generated through inference of p-site to enable visualization of RPF of all sizes. Unless stated otherwise, RNA-Seq reads of all lengths were smoothed with a 15-nt-long sliding window.

Sequence logos were produced using the R library *ggseqlogo*[99]. Shine–Dalgarno sequence strength was calculated using the Ribosome Binding Site Calculator version 1.0[100], and the anti-Shine–Dalgarno sequence in the program was modified for the analysis of *Listeria* sequences. CAI was calculated by EMBOSS CAI[101] and either the Esalty.cut or codon usage tables generated with EMBOSS CUSP. Minimal free energy (MFE) and base-pairing of RNA secondary structures was predicted by RNAFold from the ViennaRNA package[102].

Unless described otherwise, $\log_{10}$ values of mRNA abundance and protein synthesis on charts are given as $\log_{10}(x + 1)$ values. *Listeria* gene ontology was determined with STRINGdb.

## Reporting summary
Further information on research design is available in the Nature Portfolio Reporting Summary linked to this article.

## Data availability
The raw and processed RiboSeq/RNASeq data generated in this study have been deposited in the ArrayExpress database under accession code E-MTAB-11527. https://www.ebi.ac.uk/ena/browser/view/PRJEB51486.

## Code availability
Software package riboSeqR[60] used in this study is opensource under BioConductor. Customised scripts are available upon request.

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

## Acknowledgements
We would like to thank Dan Portnoy, University of Berkeley, for the generous gift of the *Listeria* 10403 S strain, and Clare Bryant and Gillian Fraser, University of Cambridge, for the generous gift of the *Salmonella* SL1344 and SJW1103 LT2-derived strains, respectively. We would also like to thank Ian Brierley, John Carr, Julian Hibberd, Nancy Standart, and Ben Luisi for discussions and comments on the manuscript. F.L. was supported by a BBSRC DTP studentship. B.Y.W.C. and O.J.B. are supported by a Medical Research Council Fellowship to B.Y.W.C. [MR/R021821/1]. J.P. is supported by a BBSRC project grant awarded to B.Y.W.C. [BB/V006096/1]. The B.Y.W.C. laboratory is supported by a Medical Research Council Fellowship [MR/R021821/1] and BBSRC project grants [BB/X001261/1, BB/V017780/1 and BB/V006096/1] and a Royal Society Research Grant [RGS\R2\192222].

## Author contributions
B.Y.W.C., O.J.B., F.L. conceived the research and designed experiments. B.Y.W.C., O.J.B., and J.P. generated RiboSeq and RNA-Seq libraries, O.J.B. performed most molecular experiments and in vivo functional assays. F.L. performed most of the bioinformatic analysis. B.Y.W.C., O.J.B., and F.L. wrote the manuscript.

## Competing interests
The authors declare no competing interests.
