## [Peer Review File · Nature Communications]

REVIEWER COMMENTS

Reviewer #1 (Remarks to the Author):

The manuscript compared the translational control mechanisms in two pathogenic bacteria: the Gram-negative bacterium *Salmonella* and the Gram-positive bacterium *Listeria*, by employing an improved ribosome profiling protocol along with parallel transcriptomics. The study suggested that *Salmonella* and *Listeria* employ different strategies to translationally regulate protein synthesis. The authors also highlight the presence of a novel translational initiation mechanism in *Listeria*.

The focus on exploring the translational control mechanisms in these two bacterial groups is highly significant. In microbiology, researchers often heavily rely on a few model organisms for investigating translation strategies, such as the Gram-negative bacterium *E. coli*, while there are likely numerous distinct regulatory aspects of gene expression that warrant further investigation given the vast diversity of microorganisms. However, the current manuscript needs significant improvement to match the title "the distinct translational landscapes of Gram-positive and Gram-negative bacteria"

Some major concerns include:

First, the differences observed in the translation behaviors of *Salmonella* and *Listeria* may not be attributed to their Gram-positive and Gram-negative classifications. These two bacterial species have undergone divergent evolutionary paths, and multiple factors could contribute to their dissimilarities. For instance, they may have different nutrient requirements, and variations in nutrient limitations have been known to influence translation strategies in bacteria (PMID: 30038306). Additionally, despite both *Salmonella* and *Listeria* being pathogenic bacteria, they exhibit discrepancies in motility and adaptability ranges. Hence, the disparities in translation regulation might be more related to the distinction between mobile and non-mobile bacteria rather than Gram-positive or Gram-negative categorization. In addition, paragraph starting at line 134 stated that the DSN treatments of these two bacteria are different, and one needs to check whether these different data processing procedures may contribute to the observed translational difference.

Second, the comparison between *Salmonella*, *Listeria*, and *E. coli* is insufficient to claim distinct translational landscapes between Gram-positive and Gram-negative bacteria. Despite that *E. coli* has been a prominent model organism for generating ribosome profiling data, substantial progress has been made in sequencing and curating Riboseq data from numerous other bacteria (PMID: 30335166), including the Gram-positive model organism *Bacillus subtilis* and the Gram-positive pathogenic bacterium *Pseudomonas aeruginosa*. A more comprehensive and systematic comparison, encompassing

all available data, is necessary to establish a definitive understanding of the translational differences between Gram-positive and Gram-negative bacteria.

Third, in terms of the novelty of this work, the finding that "SD strength is not a direct hallmark of translational efficiency" may not be considered a groundbreaking discovery. Even in the case of *E. coli*, it has been observed that translation efficiency can vary across different nutrient limitations for the same gene (PMID: 36264977). Additionally, the role of Shine-Dalgarno (SD) sequences in regulating translational efficiency has been a subject of controversy (PMID: 25636133). While the potential novel translation initiation mechanism (OUP) identified in this study warrants further attention, it should be approached with caution and examined in comparison to other bacteria to determine its generality and broader implications.

Additionally, translation efficiency (TE), the most important concept in this work, has not been defined clearly in the main text. The definition provided at line 111, stating it as the "number of ribosome-protected fragments per mRNA," is vague and lacks crucial details about whether the count has been appropriately normalized by gene length or total counts, and how was the small value in mRNA count handled. If normalization was not appropriately performed in calculating TE, many artificial correlations might be generated. Clear and explicit clarification regarding TE calculation and normalization procedures is necessary to ensure the validity and accuracy of the observed results.

Other minor points include:

It would be beneficial to establish a quantitative association between stoichiometry, mRNA abundance, and translation efficiency (TE). This could be achieved by employing correlation coefficients or multiple linear regression analyses that incorporate stoichiometry, TE, and RPKM (Reads Per Kilobase Million) values for protein complexes. By systematically assessing the coupling of these variables, a clearer understanding of their relationships can be obtained.

The determination of the two groups of ribosome genes mentioned at lines 297 to 299 should be explained more clearly in the paper.

The discovery of OUP is indeed noteworthy and deserves attention. Conducting a systematic survey to analyze the statistics of genes with OUP initiation, such as their COG (Clusters of Orthologous Groups) category, would be highly beneficial to relate the observed difference into biological explanations.

Reviewer #2 (Remarks to the Author):

The manuscript by Bryant et al., reports a study on comparative ribosome profiling results from pathogenic Gram-negative Salmonella and Gram-positive Listeria. Authors provide some evidence that translation might be regulated differently between two organisms, increased translational efficiency for Lys-codons in the beginning of the mRNAs and presence of “out-of frame upstream peaks” (OUPs) for Listeria as a possibly new translational control mechanism associated with 15% of Listeria transcripts. The overall data as well as comparative ribosome profiling data and analyses present interesting set of observations however additional analyses and discussions are needed, as well as whether OUPs indeed represent new translational control mechanism or artifact. Multiple figures are misquoted through the manuscript and the figure legends are not correct. Detailed concerns are in more details below.

Detailed questions and comments:

- Is there a concern that single DSN treatment works well for Salmonella but not for Listeria? Is this associated with GC (or AT richness) of the genomes, overall lower complexity of sequences and could multiple rounds of DSN treatment skew results of analyses?
- Authors argue (in line 158) that metagene analysis indicates pausing at translation initiation regardless of use of different nucleases (Fig1A, S2B). This is correct for Salmonella but not really seen in Listeria. Could different susceptibility of these bacteria to chloramphenicol be a reason for this? Listeria is much less susceptible to chloramphenicol.
- It is not clear from figure S2C why S7 MNase-treated Ribo-Seq tends to overestimate protein synthesis (line 166-167)
- Authors argue that the sequences in Listeria with OUPs have slightly strong SD sequence – what does this really mean sequence wise and is it a significant difference? Figure 1B is misannotated for showing this difference and I couldn't find corresponding figure for this claim (line 186).
- If stronger SDs in OUP-containing sequences increase the TE of these genes is this different than what is observed in other bacteria including Salmonella, the stronger the SD higher efficiency of initiation and TE?
- Could OUP reads represent longer reads associated with initiation or “stronger SD binding” as reported for longer SD-associated reads in elongating ribosomes in study by O'Connor et al., 2013 (Bioinformatics)
- While it is obvious how 24, 25, 28 and 29 nucleotide reads are more phased than 26 in Salmonella, it is not clear why 27 are not. They look similar to 28 and 29 nucleotide reads. What about Listeria reads?

- In the section on stoichiometric production through coordinated translation of protein complexes with virulence - atpE removal is associated with a bias during the ligation. How many other mRNAs are associated with such bias and what is the bias?

- There are more determinants of efficient protein synthesis besides SD sequence strength and CAI. The manuscripts of Verma et al., 2019(Nat Comm), Moreira et al., 2019 (RNA Biol), Saito et al., 2020 (eLife), Osterman et al., 2020 (NAR), Umemoto et al., 2023 (NAR), Nagao et al., 2023 (Nat Comm) point out to analyses of SD, mRNA structure, mRNA and protein sequence on efficiency of the translation initiation and elongation or overall translation in bacteria. Some of these studies (Verma et al., Moreira et al., Osterman et al.) indicate as well amino acids (Lys-Lys pairs) or as well AU-richness (AAA-AAA) as determining factors in translation efficiency in E.coli, E. asburiae and K. oxytoca. This whole section is also not cited in section about sequence determinants (lines 364 -435)

- Is correlation of TE with SD (Fig 3F) in blue group of Listeria really sensitive to SD strength and whether the blue and red group in both bacteria represent different ribosome genes or they are the same? I would expect that two groups are using same proteins due to conservation of sequences.

- Fig 4 H-J legend is not correct (or confusing) so it is hard to assess what is in each panel which makes it hard to follow discussion of the peak sizes described in lines 350 -362. Are mentioned longer reads in initiation for OUP-genes associated with the “strong SD in these genes” and similar to observation by O’Connor et al., 2013.

- The whole section on sequence context surrounding the start codon is missing references to previous work associated with this part of mRNA or protein sequence and studies in E.coli and other bacteria (mentioned above)

Line 30 – there is a full stop missing.

Line 186 – quotes figure 1B for stronger SD sequence in Listeria. There is no such representation in figure 1B

Line 463 – there is no Figure S4D

Reviewer #3 (Remarks to the Author):

In their study Bryant et al. employ ribosome profiling (Ribo-seq) to address differences in translation regulation between two bacteria, Salmonella typhimurium and Listeria monocytogenes using improved Ribo-seq (i.e. with a mixture of two enzymes, RNase I and MNase, to generate the footprints). By analyzing the distribution of footprints the authors concluded substantial differences in translation

initiation in both bacteria that do not depend on the strength of Shine-Dalgarno (SD) sequence. Extant literature suggests that indeed gram-positive and gram-negative bacteria show distinct patterns of translational regulation, with the Shine-Dalgarno sequence not being the major determinant of translation efficiency. Thus, this conclusion aligns well with the literature. Furthermore, tandem Lys (AAA-AAA) codons are enriched in the immediate vicinity of the start codon and suggest that unstructured site surrounding the start site facilitates expression.

Two major problems I see with this study: (I) the way the analyses are performed do not substantiate the conclusions, and (II) translation regulatory features are concluded without considering important parameters. Here some elaboration on the problems:

(I) To understand the role of the initiation vs elongation, the data need a calibration, i.e. each read length should be aligned to the A or P site. Since the authors use combination of two nucleases, the reads should be calibrated to both 3' and 5' ends. Only such calibration would allow for determination of codon usage, elongation regulation or extract regulatory sequences with codon-specific identities (including also the tandem lysines). The paper is mostly built on single cases. The richness of the Ribo-seq and the accompanying RNA-seq should be fully exploited. A very broad distribution of read lengths has been observed; the fact that different read lengths can represent different ribosomal states should be addressed. For this, the reads should be grouped by length, calibrated to the A site (or P site) for each length calibrated and through the cumulative plots show the validity for a large amount of transcripts. The data should be accompanied with appropriate statistical analysis.

(II) For the secondary structure analysis predictive secondary analysis are less informative. The strength of the SD is not defined by the propensity to form secondary structure, but rather by the strength of interactions with the anti-SD. Thus, the minimum hybridization energy is a more appropriate measure when assessing the strength of the SD. This measure will allow grouping all transcripts in categories and determine translation efficiency for each group.

(III) Bacterial genes are organized in operons with some overlapping genes making impossible to assign the corresponding reads in the vicinity of start codons (or end of the upstream transcripts). Thus, the different gene groups (transcript in operons non-overlapping, overlapping and independently expressed, non-operon) should be analyzed separately. This may alter some conclusions in the manuscript as the 'out-of-frame' upstream peaks. It should be shown that those are not from transcripts that are overlapping or in a close proximity with each other (i.e. distance less than a ribosome size). Furthermore, such separation of the transcripts in groups may explain the pattern seen in Fig. 3F. Many genes within one operon share SD, or through the overlap bypass the requirement for a SD. Thus, it is important to divide the transcripts based on their architecture and separately look into each group.

Additional important remarks:

- Reading the through description of the ribosome profiling in the introduction, one would expect a methodological paper. Despite Ribo-seq being relatively complex technology, it is fairly well established so that such lengthy introduction on its principle is not necessary. Far more important is to include all relevant literature, including seminal papers about the lack of effect of SD, enrichment of A/T-rich codons downstream of the start codon.

- Growth curves should be included to show that the comparison is done at the same growth phase for both organisms. Ideally, both should be in the mid exponential phase.

- The depth of the libraries after mapping to the genome should be shown.

- The enrichment of Asn and Lys at position 2 and 3 downstream of the AUG codon may not feel unexpected, if one would consider previous studies suggesting enrichment of A/T rich codons at these positions across various bacterial strains (including the tested organisms in this study; PMID: 24072823; PMID 23774758,).

In sum, the paper presents a quite rich deep-seq data analysis, but the analyses feel premature and inappropriate to substantiate the conclusions.

Response to Reviewers' Comments

We would like to thank the reviewers for their constructive comments, which we have attempted to address in full, as far as we can. We think that the changes have considerably strengthened our manuscript. Below, we give a detailed point-by-point account of the changes we have made in response to the comments. In the uploaded copy of our revised manuscript we have highlighted the major changes requested by reviewers in yellow.

We believe we have exhausted all approaches to answer the reviewers' comments but welcome any further suggestions.

Reviewer #1 (Remarks to the Author):

The manuscript compared the translational control mechanisms in two pathogenic bacteria: the Gram-negative bacterium *Salmonella* and the Gram-positive bacterium *Listeria*, by employing an improved ribosome profiling protocol along with parallel transcriptomics. The study suggested that *Salmonella* and *Listeria* employ different strategies to translationally regulate protein synthesis. The authors also highlight the presence of a novel translational initiation mechanism in *Listeria*.

The focus on exploring the translational control mechanisms in these two bacterial groups is highly significant. In microbiology, researchers often heavily rely on a few model organisms for investigating translation strategies, such as the Gram-negative bacterium *E. coli*, while there are likely numerous distinct regulatory aspects of gene expression that warrant further investigation given the vast diversity of microorganisms. However, the current manuscript needs significant improvement to match the title "the distinct translational landscapes of Gram-positive and Gram-negative bacteria"

Some major concerns include:

1a. First, the differences observed in the translation behaviors of *Salmonella* and *Listeria* may not be attributed to their Gram-positive and Gram-negative classifications. These two bacterial species have undergone divergent evolutionary paths, and multiple factors could contribute to their dissimilarities. For instance, they may have different nutrient requirements, and variations in nutrient limitations have been known to influence translation strategies in bacteria (PMID: 30038306). Additionally, despite both *Salmonella* and *Listeria* being pathogenic bacteria, they exhibit discrepancies in motility and adaptability ranges. Hence, the disparities in translation regulation might be more related to the distinction between mobile and non-mobile bacteria rather than Gram-positive or Gram-negative categorization.

Thank you for bringing this to our attention, we have now included additional text on this in the introduction and discussion sections of our manuscript (see lines 55-61). We have also modified our manuscript title based on the above comments. We have used both the gram-positive *Listeria* and gram-negative *Salmonella* as representatives of gram-positive and gram-negative bacteria and we believe this has set the groundwork for future work to accurately study translation control in other pathogenic bacterial species that belong to gram-positive and gram-negative classifications such as *Pseudomonas aeruginosa*, *Klebsiella pneumoniae*, *Staphylococcus aureus* and *Streptococcus pneumoniae*.

1b. In addition, paragraph starting at line 134 stated that the DSN treatments of these two bacteria are different, and one needs to check whether these different data processing procedures may contribute to the observed translational difference.

We have compared the effect of different numbers of DSN treatments (1× DSN treatment vs 3× DSN treatment) on the normalised read count for each gene. The two *Listeria* libraries analysed originate from the same sample. The normalised read counts in the two libraries are consistent, which indicates that additional rounds of DSN treatment did not meaningfully skew the quantification of protein synthesis (figure R1, also included in the revised figure panel as Figure S2E).

Figure R1: Comparison of normalised RPF count in *Listeria* libraries treated 1× or 3× with DSN. Only genes with sufficient read counts present in both libraries are charted. Read count where the count difference was greater than three-fold are highlighted in red, which is only 1 gene. Ribosomal genes are charted in yellow.

Figure 1B: Library composition of RiboSeq libraries from *Listeria* (10403S) generated with RNase I that were subjected to oligonucleotide-based rRNA subtraction, followed by one or three rounds of DSN treatment. Three rounds of DSN treatment significantly enriched mRNA reads over 1 round of DSN treatment.

1c. Second, the comparison between *Salmonella*, *Listeria*, and *E. coli* is insufficient to claim distinct translational landscapes between Gram-positive and Gram-negative bacteria. Despite that *E. coli* has been a prominent model organism for generating ribosome profiling data, substantial progress has been made in sequencing and curating RiboSeq data from numerous other bacteria (PMID: 30335166), including the Gram-positive model organism *Bacillus subtilis* and the Gram-positive pathogenic bacterium *Pseudomonas aeruginosa*. A more comprehensive and systematic comparison, encompassing all available data, is necessary to establish a definitive understanding of the translational differences between Gram-positive and Gram-negative bacteria.

Thank you for pointing this out and it highlights the need to perform these analyses with other bacterial systems using our optimised ribosome profiling protocol. The currently available bacterial Ribosome profiling datasets (including for *Bacillus subtilis* and *Pseudomonas aeruginosa*) do not have appropriately clear phasing to achieve the type of analysis that we could perform with our data. Phased data reflects triplet decoding nature of the ribosome and therefore enables accurate assignment of the P-site codon position for all reads, which is a critical step in accurate quantitative analysis of translation properties at the codon level. Inaccurate assignment of the A- and P-sites introduces random or systematic errors that weakens biological signals observed. This inaccuracy hinders the ability to quantitatively understand translation at single nucleotide resolution. Our optimised protocol (by optimising translation inhibition, nuclease and digestion conditions and the depletion of contaminating ribosomal RNA) allowed us to reproducibly obtain bacterial ribosome profiling data with high levels of phasing. We are restricted to comparing ribosome profiling data from only two organisms (*Listeria* and *Salmonella*) as these have sufficient phasing to allow accurate A- and P-site assignment. We have now modified the title to reflect this.

1d. Third, in terms of the novelty of this work, the finding that "SD strength is not a direct hallmark of translational efficiency" may not be considered a groundbreaking discovery. Even in the case of *E. coli*, it has been observed that translation efficiency can vary across different nutrient limitations for the same gene (PMID: 36264977). Additionally, the role of Shine-Dalgarno (SD) sequences in regulating translational efficiency has been a subject of controversy (PMID: 25636133). While the potential novel translation initiation mechanism (OUP) identified in this study warrants further attention, it should be approached with caution and examined in comparison to other bacteria to determine its generality and broader implications.

We fully agree with this comment, we did not think this was groundbreaking but we thought it was necessary to point out the relationship between SD strength and translation efficiency as our data enables us to do this accurately. There is a lot of controversy in the literature about the contribution that Shine-Dalgarno strength plays in controlling translation efficiency. We therefore set out to examine this controversial relationship using our high-resolution data. As our data contains a high level of phasing, we can accurately determine translation efficiency (TE) which allows us to quantify TE with confidence. We found that the correlation between Shine-Dalgarno strength and translation efficiency is not broadly applied to all genes but is an important hallmark for certain genes, we have highlighted this in the discussion section. A good example worth noting is the *atp* operon which contains genes encoding components of the F_1F_0 -ATPase complex (Fig.S8) and the SufBCD complex (Fig. S9) where we see a positive correlation between Shine-Dalgarno strength and TE, whereas when we examine the correlation between Shine-Dalgarno strength and TE at the global level we see a much weaker correlation (Fig. s12). We agree with the reviewer that it would be crucial to examine and compare this correlation in other bacteria in future. While profiling other bacteria is out of scope of this paper, it is something we would like to venture into in the future for the scientific community.

1e. Additionally, translation efficiency (TE), the most important concept in this work, has not been defined clearly in the main text. The definition provided at line 111, stating it as the "number of ribosome-protected fragments per mRNA," is vague and lacks crucial details about whether the count has been appropriately normalized by gene length or total counts, and how was the small value in mRNA count handled. If normalization was not appropriately performed in calculating TE, many artificial correlations might be generated. Clear and explicit clarification regarding TE calculation and normalization procedures is necessary to ensure the validity and accuracy of the observed results.

Thank you for bringing this to our attention, we have now addressed this in the manuscript (Main manuscript: lines 118-122 and extended material: Bioinformatics section). The transcript abundance is quantified as the number of RNA-Seq reads per kilobase of CDS per million RNA-Seq reads (RPKM), and the protein synthesis is quantified as the number of RPFs per kilobase of CDS per million phased RPFs. Translational efficiency of a gene is the ratio of phased RPFs mapped to the coding region normalized to the total number of phased RPFs in all coding regions and RNA-Seq reads mapped to the coding region normalized to the total number of RNA-Seq reads mapped to all coding regions.

Normalisation methods for intersample comparisons, such as TMM, GeTMM, or median-of-ratios, were not used because no intersample comparison was performed. All interlibrary comparisons in this article are performed on libraries that originate from the same samples.

Other minor points include:

1f. It would be beneficial to establish a quantitative association between stoichiometry, mRNA abundance, and translation efficiency (TE). This could be achieved by employing correlation coefficients or multiple linear regression analyses that incorporate stoichiometry, TE, and RPKM (Reads Per Kilobase Million) values for protein complexes. By systematically assessing the coupling of these variables, a clearer understanding of their relationships can be obtained.

Thank you for this, we did not make this very clear and we have now described this in lines 118-122, and Figures 2, 3, S7, S12, S13, S18, S19. The correlation coefficients and p-values are embedded in the figures where possible – we were not able to perform these analyses for charts containing too few data points and hence we have not performed these analyses for those examples and instead show the data points as examples.

1g. The determination of the two groups of ribosome genes mentioned at lines 297 to 299 should be explained more clearly in the paper.

Thank you for bringing this to our attention, we have now addressed this in the manuscript (lines 309-310). We stratified the data points based on whether they lay above or below $y = mx + c$ in the Shine-Dalgarno vs translation efficiency plots. Data points above $y = mx + c$ are highlighted in red and are sensitive to Shine-Dalgarno strength. Data points below $y = mx + c$ are highlighted in blue and are not sensitive to Shine-Dalgarno strength.

1h. The discovery of OUP is indeed noteworthy and deserves attention. Conducting a systematic survey to analyze the statistics of genes with OUP initiation, such as their COG (Clusters of Orthologous Groups) category, would be highly beneficial to relate the observed difference into biological explanations.

We performed these analyses and we have included this data in Figure 4G (right).

We have also generated a Euler diagram showing the overlap of selected gene ontology groups in *Listeria* OUP genes. This diagram is located in Figure S20E (left).

Reviewer #2 (Remarks to the Author):

The manuscript by Bryant et al., reports a study on comparative ribosome profiling results from pathogenic Gram-negative *Salmonella* and Gram-positive *Listeria*. Authors provide some evidence that translation might be regulated differently between two organisms, increased translational efficiency for Lys-codons in the beginning of the mRNAs and presence of “out-of frame upstream peaks” (OUPs) for *Listeria* as a possibly new translational control mechanism associated with 15% of *Listeria* transcripts. The overall data as well as comparative ribosome profiling data and analyses present interesting set of observations however additional analyses and discussions are needed, as well as whether OUPs indeed represent new translational control mechanism or artifact. Multiple figures are misquoted through the manuscript and the figure legends are not correct. Detailed concerns are in more details below.

Detailed questions and comments:

2a. Is there a concern that single DSN treatment works well for *Salmonella* but not for *Listeria*? Is this associated with GC (or AT richness) of the genomes, overall lower complexity of sequences and could multiple rounds of DSN treatment skew results of analyses?

The GC or AT richness does not affect the DSN treatment but rather the number of homologous fragments that are present in the sample. Highly abundant fragments (such as ribosomal RNA) are more likely to anneal and form double stranded DNA that are sensitive to digestion by DSN. As there are very few mRNA reads, DSN treatment has no significant effect on the level of even the most abundant mRNA fragments (see the response to reviewer 1 (point 1b, and Figure R1 below) and the analysis we have performed using our data (Fig. S2E)). We have performed DSN treatment in a range of organisms, from algae (68% GC rich) to mice (42% GC rich) and see no significant affect and we do not see a correlation with the effectiveness of the DSN treatment (Chung *et al.*, RNA, 2015). However, we would like to highlight that the main reason for three rounds of DSN treatment for *Listeria* was due to the sub-par oligo-based depletion prior to DSN treatment. We have found that the commercial oligo based rRNA depletion is less efficient for *Listeria* than for *Salmonella* which would account for the need to apply more rounds of DSN treatment to deplete rRNA from the *Listeria* sample and enrich for mRNA reads.

Figure R1: Comparison of normalised RPF count in *Listeria* libraries treated 1x or 3x with DSN. Only genes with sufficient read counts present in both libraries are charted. Read count where the count difference was greater than three-fold are highlighted in red, which is only 1 gene. Ribosomal genes are charted in yellow.

2b. Authors argue (in line 158) that metagene analysis indicates pausing at translation initiation regardless of use of different nucleases (Fig1A, S2B). This is correct for *Salmonella* but not really seen in *Listeria*. Could different susceptibility of these bacteria to chloramphenicol be a reason for this? *Listeria* is much less susceptible to chloramphenicol.

We are currently unsure of the reason for this, although, we would not be able to capture OUP, a novel rate-limiting step of translational initiation if *Listeria* is less susceptible to chloramphenicol. We did however ensure that the concentration of chloramphenicol we used to inhibit translation in *Listeria* is above the reported MIC of chloramphenicol for *Listeria monocytogenes* (reported to be between 8-16 µg.ml, doi: 10.1128/aac.43.9.2103). The chloramphenicol concentrations we used to initially inhibit translation was 300 µg.ml which is well above the MIC. Cells treated with chloramphenicol were rapidly cooled and frozen to prevent further translation. We also maintained this concentration of chloramphenicol in all the buffers until we had generated ribosome footprints. It is also worth noting that the chloramphenicol binding site in the *Listeria* ribosome is predicted to be conserved with that of *E. coli* and *Salmonella* and so upon generation of the extracts, the ribosomes will be fully accessible to the chloramphenicol in the same way that the *Salmonella* ribosomes are in the *Salmonella* extracts.

2C. It is not clear from figure S2C why S7 MNase-treated Ribo-Seq tends to overestimate protein synthesis (line 166-167)

The reason for this is that we performed the analyses for this chart with all reads. This was necessary because there is no phasing in the data generated using MNase, whereas the data generated using RNase I is highly phased. Data generated with MNase does not allow us to distinguish *bona fide* footprints from RNA contamination, whereas data generated with RNase I data can as we have highly phased reads, and we can determine the P-site. Therefore, overestimation of protein synthesis can be explained by a combination of both RNA contamination and *bona fide* RPF's contributing to RPF read counts, an issue that will occur in all ribosome profiling that do not contain phasing. An important issue which we have overcome with our optimised ribosome profiling method. The overestimation of protein synthesis in libraries generated with MNase I is compounded for genes with a lower ribosome density on the transcript due to a greater contribution of RNA contamination to the signal. Despite having highly phased data when using RNase I, we excluded non-phased reads (the small amount present in the data) from the data for analyses of all other charts in the manuscript.

We do however apologise an oversight on the original figure S2C, we also included genes with fewer than 50 RPF counts. We have now replaced figure S2C with figure below (now figure S2D).

2d. Authors argue that the sequences in *Listeria* with OUPs have slightly strong SD sequence – what does this really mean sequence wise and is it a significant difference? Figure 1B is misannotated for showing this difference and I couldn't find corresponding figure for this claim (line 186).

We looked at the difference in Shine-Dalgarno strength for genes with and without OUPs and found that there is only a marginal difference in Shine-Dalgarno strength. We show that OUP is not due to stronger Shine-Dalgarno strength, nor due to the distance between the Shine-Dalgarno and start codon of OUP vs non-OUP genes. We believed it is important to present this information in the manuscript. To address the second point, we apologise for this mistake, and we have now corrected this in the manuscript.

2e. If stronger SDs in OUP-containing sequences increase the TE of these genes is this different than what is observed in other bacteria including *Salmonella*, the stronger the SD higher efficiency of initiation and TE?

There is only a small difference in Shine-Dalgarno strength of genes with and without a prominent OUP, which is unlikely to explain the marked difference in translational efficiencies. We also found that the distance between the Shine-Dalgarno sequence and the start codon does not differ between OUP and non-OUP genes, suggesting that the spacer is not a determinant of the OUP. We apologise for not making this clearer in the manuscript and we have modified the manuscript accordingly (lines 196-197, 352-355).

2f. Could OUP reads represent longer reads associated with initiation or “stronger SD binding” as reported for longer SD-associated reads in elongating ribosomes in study by O'Connor et al., 2013 (Bioinformatics)

The length distribution of OUP reads is similar to the one of elongating reads. (See figure 4I, left). However, it may still be the case that OUP reads originate from longer initiating reads, which were cloven at a position further upstream than ordinarily seen with elongating reads, as we hypothesize in the discussion. Further credence is lent to this hypothesis by the fact that eukaryotic ribosomes produce shorter footprints when

the A-site is unoccupied by tRNA and is more accessible to RNase I (Wu *et al.* 2019, 10.1016/j.molcel.2018.12.009), and the ribosomal A-site is vacant during translational initiation. Digestion further upstream distorts the calculation of P-site position.

The reason why we do not see a prominent OUP in *Salmonella* may be a different ribosomal structure, which does not allow this access.

Listeria tuf gene. 24–25-nt-long reads. P-site position on x-axis. No prominent OUP.

This, however, does not explain why we only see OUP in a subset of genes with a range of translational efficiencies, albeit more efficiently translated on average. (For example, *tuf* is translated at a very high level, but it does not have a prominent OUP. left) It is important to note that we do not see longer or more abundant initiation reads that map to the rest of the genes. (See figure 4I above)

Thank you for bringing the work by O'Connor et al., 2013 to our attention, we have added this to the list of other previous work that complements our analyses (Line 371-373). Most importantly, the analysis by O'Connor et al., 2013 was performed with ribosome profiling data generated with MNase I, which is not phased and

therefore it is not possible to differentiate *bona fide* reads from RNA contamination which would explain the broader footprint distribution observed by O'Connor et al., 2013. However, we do agree with the review that we must acknowledge this paper and we have now cited this paper in our manuscript.

2g. While it is obvious how 24, 25, 28 and 29 nucleotide reads are more phased than 26 in *Salmonella*, it is not clear why 27 are not. They look similar to 28 and 29 nucleotide reads. What about *Listeria* reads?

The phasing of 27-nt-long reads in *Salmonella* SL1344 is still very clear. (See figure S3A, left) We did not include longer reads from the *Listeria* Ribo-Seq library because the abundance of such reads was too low. (See figure S3, right). Whilst we did size select ribosome footprints at between 18–34 nt, we are reluctant to say that *Listeria*

does not have nucleotides reads of 27–29 unless we repeat the experiment on a larger size range.

2h. In the section on stoichiometric production through coordinated translation of protein complexes with virulence - *atpE* removal is associated with a bias during the ligation. How many other mRNAs are associated with such bias and what is the bias?

Ribosome profiling is full of biases and there are many reasons for bias (e.g. 3' adaptor ligation, 5' adaptor ligation, PCR amplification etc...). As our data is highly phased we can directly visualise translation at the codon level and identify unusual behaviour such as that seen in the *atpE* gene and exclude these biases. We performed RNA-seq in parallel with parallel size selection and in the case of *atpE* we saw the same bias. We have not performed a global search for bias but all examples of genes that we used in the text did not contain obvious bias other than *atpE*.

2i. There are more determinants of efficient protein synthesis besides SD sequence strength and CAI. The manuscripts of Verma et al., 2019 (Nat Comm), Moreira et al., 2019 (RNA Biol), Saito et al., 2020 (eLife), Osterman et al., 2020 (NAR), Umemoto et al., 2023 (NAR), Nagao et al., 2023 (Nat Comm) point out to analyses of SD, mRNA structure, mRNA and protein sequence on efficiency of the translation initiation and elongation or overall translation in bacteria. Some of these studies (Verma et al., Moreira et al., Osterman et al.) indicate as well amino acids (Lys-Lys pairs) or as well AU-richness (AAA-AAA) as determining factors in translation efficiency in *E. coli*, *E. asburiae* and *K. oxytoca*. This whole section is also not cited in section about sequence determinants (lines 364 -453)

We apologise for this mistake; we have amended the text and have included these citations (line 390 and lines 411-416).

2j. Is correlation of TE with SD (Fig 3F) in blue group of *Listeria* really sensitive to SD strength and whether the blue and red group in both bacteria represent different ribosome genes or they are the same? I would expect that two groups are using same proteins due to conservation of sequences.

Yes, we believe the blue group of *Listeria* is less sensitive to SD strength (See also comment 1g from reviewer 1). We have now coloured the red genes in *Salmonella* on the *Listeria* ribosome chart (left) which suggests that the group of *Salmonella* ribosome genes that are more sensitive to Shine-Dalgarno strength do not follow the same trend in *Listeria*. This indicates that translation efficiency of these genes are regulated differently in both *Salmonella* and *Listeria*.

Shine-Dalgarno strength of *Listeria* ribosomal protein genes plotted as a function of translation efficiency. *Salmonella* ribosomal genes that are sensitive to Shine-Dalgarno strength (Fig. 3F) are highlighted on the *Listeria* plot above (red) showing that the ribosomal genes do not display the same trend in *Salmonella* and *Listeria*.

2k. Fig 4 H-J legend is not correct (or confusing) so it is hard to assess what is in each panel which makes it hard to follow discussion of the peak sizes described in lines 350 -362. Are mentioned longer reads in initiation for OUP-genes associated with the “strong SD in these genes” and similar to observation by O’Connor et al., 2013.

We apologise, we have now modified the figure legend for Figure 4 H-J. We are cautious as to whether there is a significant difference in Shine-Dalgarno strength between genes with and without OUP’s, it would however be interesting to see if OUPs are present in *Bacillus subtilis* when highly phased ribosome profiling in *Bacillus* becomes available.

2l. - The whole section on sequence context surrounding the start codon is missing references to previous work associated with this part of mRNA or protein sequence and studies in *E.coli* and other bacteria (mentioned above)

Thank you for bringing this to our attention, have now included multiple references in light of this.

2m. Line 30 – there is a full stop missing.

Thank you for pointing this out, we have corrected this in the text.

2n. Line 186 – quotes figure 1B for stronger SD sequence in *Listeria*. There is no such representation in figure 1B

We apologise, we have now corrected this in the manuscript to Figure 4E.

2o. Line 463 – there is no Figure S4D

We apologise, we have now corrected this in the manuscript to Figure S4A.

Reviewer #3 (Remarks to the Author):

3a. In their study Bryant et al. employ ribosome profiling (Ribo-seq) to address differences in translation regulation between two bacteria, *Salmonella typhimurium* and *Listeria monocytogenes* using improved Ribo-seq (i.e. with a mixture of two enzymes, RNase I and MNase, to generate the footprints). By analyzing the distribution of footprints the authors concluded substantial differences in translation initiation in both bacteria that do not depend on the strength of Shine-Dalgarno (SD) sequence. Extant literature suggests that indeed gram-positive and gram-negative bacteria show distinct patterns of translational regulation, with the Shine-Dalgarno sequence not being the major determinant of translation efficiency. Thus, this conclusion aligns well with the literature. Furthermore, tandem Lys (AAA-AAA) codons are enriched in the immediate vicinity of the start codon and suggest that unstructured site surrounding the start site facilitates expression.

Thank you for this comment, we agree that the enrichment of tandem lysines at the vicinity of the start codon suggests reduced RNA structure at the start site. It is worth noting that not all efficiently translated transcripts have unstructured start sites, suggesting multiple mechanisms exist to regulate efficient translation efficiency. We have generated a Venn diagram (left) of efficiently translated transcripts with unstructured translation start site, high SD strength, AAA-AAA codons immediately following the start codon or none of above.

3b. Two major problems I see with this study: (I) the way the analyses are performed do not substantiate the conclusions, and (II) translation regulatory features are concluded without considering important parameters. Here some elaboration on the problems:

(I) To understand the role of the initiation vs elongation, the data need a calibration, i.e. each read length should be aligned to the A or P site. Since the authors use combination of two nucleases, the reads should be calibrated to both 3' and 5' ends. Only such calibration would allow for determination of codon usage, elongation regulation or extract regulatory sequences with codon-specific identities (including also the tandem lysines). The paper is mostly built on single cases. The richness of the Ribo-seq and the accompanying RNA-seq should be fully exploited. A very broad distribution of read lengths has been observed; the fact that different read lengths can represent different ribosomal states should be addressed. For this, the reads should be grouped by length, calibrated to the A site (or P site) for each length calibrated and through the cumulative plots show the validity for a large amount of transcripts. The data should be accompanied with appropriate statistical analysis.

We apologise that we did not make this more clear in the manuscript. We can address both these points. For point (I) As our data is highly phased, we analysed the data by referring to the P-site codon for all size classes and only used the phased reads for this (the majority of reads are phased). We did not use a combination of nucleases to generate our data. We used MNase I to confirm

previous findings that the generation of reads with phasing using MNase I is not possible. We then used only RNase I to generate phased data using our optimised method. As we refer to the P-site our data is calibrated and therefore we could perform the analyses we performed and determine codon usage, elongation regulation or extract regulatory sequences with codon-specific identities.

To address the second point in the above comment, we have performed these analyses and have grouped reads by length, calibrated to the P-site and performed the analyses (Fig. S3A, S3B and S4). We have determined the P-site positions from RPF reads (see figure S4A), and these positions were used in constructing read maps. The distance of the P-site position from the 5' end does vary with the length of the read, and that has been considered. In gene expression analysis, only phased RPF reads were considered in the calculations.

3c. For the secondary structure analysis predictive secondary analysis are less informative. The strength of the SD is not defined by the propensity to form secondary structure, but rather by the strength of interactions with the anti-SD. Thus, the minimum hybridization energy is a more appropriate measure when assessing the strength of the SD. This measure will allow grouping all transcripts in categories and determine translation efficiency for each group.

Thank you for pointing this out, we did not make this clear in the text (see lines 318-320 and extended materials section). We have performed both of the analyses described in your comment. The predictive secondary analyses were performed without considering the Shine-Dalgarno sequence (Figure 4A), whereas Shine-Dalgarno strength was calculated using Salis's RBS calculator (Salis *et al.* 2009, Nature Biotechnology), which considers both the RNA structure of the transcript and the strength of ribosome-mRNA interaction.

3d. Bacterial genes are organized in operons with some overlapping genes making impossible to assign the corresponding reads in the vicinity of start codons (or end of the upstream transcripts). Thus, the different gene groups (transcript in operons non-overlapping, overlapping and independently expressed, non-operon) should be analyzed separately. This may alter some conclusions in the manuscript as the 'out-of-frame' upstream peaks. It should be shown that those are not from transcripts that are overlapping or in a close proximity with each other (i.e. distance less than a ribosome size). Furthermore, such separation of the transcripts in groups may explain the pattern seen in Fig. 3F. Many genes within one operon share SD, or through the overlap bypass the requirement for a SD. Thus, it is important to divide the transcripts based on their architecture and separately look into each group.

Our data is highly phased which allows us to accurately determine reading frame and enables clear classification of reads allowing us to overcome the first issue mentioned above for adjacent overlapping genes that are in a different frame. It is however difficult to determine the exact range of bacterial transcripts and which genes are co-transcribed from the genome. Prediction of promoters, terminators, and RNA-

Seq data may provide further information, but the reliability of such predictions is uncertain. The situation is further complicated by the existence of complex operons, that is operons with multiple promoters or terminators, which yield transcripts that vary in their gene composition (e.g. *flgAMN/flgMN* or *fljB/fljBA* in *Salmonella*). It is therefore challenging to determine whether OUP's are in monocistronic operons or at the 5' end of a polycistronic transcript to determine whether OUP's are from an overlapping upstream peak. However, there is a lack of reads in the 5' and 3' end of metatranslatome plots generated from OUP containing genes when compared to metatranslatome

plots generated from non-OUP genes suggesting that a greater proportion of genes containing an OUP are monocistronic. Non-OUP genes contain phased 3'UTR reads which are derived from either overlapping an ORF or an ORF immediately downstream (Figure 4C, and above).

In addition, visualisation of *rpsO* (Figure 4D, left) shows the OUP peak but does not show any reads upstream of the OUP peak, therefore confirming that the OUP peak is not due to the presence of a non-overlapping and independently expressed ORF upstream from the OUP containing gene.

3e. Additional important remarks:

- Reading the through description of the ribosome profiling in the introduction, one would expect a methodological paper. Despite Ribo-seq being relatively complex technology, it is fairly well established so that such lengthy introduction on its principle is not necessary. Far more important is to include all relevant literature, including seminal papers about the lack of effect of SD, enrichment of A/T-rich codons downstream of the start codon.

Thank you for pointing this out, as you mentioned above, Ribo-Seq is a relatively complex technology and is well established in eukaryotic systems where phasing is readily achievable. Ribo-seq in wild type pathogenic bacteria is more challenging due to a lack of phasing due to the reasons discussed in the manuscript. We heavily optimised the protocol to overcome these challenges and we are now able to reproducibly obtain ribosome profiling data with high levels of phasing. We felt the need to discuss the issues preventing the generation phased ribosome profiling data as it was critical to overcome these obstacles to allow us to obtain highly-phased data which we further discuss in the first results section. We had discussed the lack of Shine-Dalgarno sequence in some bacterial species and also cite seminal papers on the role of A/U-rich codons on translation. In light of this comment, we have included additional text and citations in the manuscript. We have also addressed this as part of a previous reviewers comment (point 2i).

3f. Growth curves should be included to show that the comparison is done at the same growth phase for both organisms. Ideally, both should be in the mid exponential phase.

Growth curves for *Salmonella* and *E.coli* have been long established in the literature and so we have used this information and performed ribosome profiling at mid-exponential phase at OD600 = 1.0.

3g. The depth of the libraries after mapping to the genome should be shown.

Thank you for this comment, we have now generated a table to show the sequencing depth of the libraries after mapping to the genome (Table 4).

	genes with mapped RPFs and RNA-Seq reads / total of annotated genes	number of phased RPFs mapping to CDS	number of mapped phased RPFs per kilobase of all CDS	number of RNA-Seq reads mapping to CDS	number of mapped RNA-Seq reads per kilobase of all CDS
Salmonella	3450/4682	19455274	4452	15936291	3601
Listeria	2155/2800	76297470	29577	57690643	22247

3h. The enrichment of Asn and Lys at position 2 and 3 downstream of the AUG codon may not feel unexpected, if one would consider previous studies suggesting enrichment of A/T rich codons at these positions across various bacterial strains (including the tested organisms in this study; PMID: 24072823; PMID 23774758,).

Yes, Lys codons were enriched but we found that Asn codons were not. We picked Asn because it is encoded by codons that start with AA (AAU, AAC), like lysine (AAA, AAG). Despite the A-richness, we did not see such a dramatic increase in reporter activity as we did with lysine-encoding codons. We

did highlight the previous observation that one lysine codon immediately adjacent to the start codon and we thank the reviewer for the additional citation. However, we found that two lysine codons have a significantly stronger effect than the previously reported single lysine codon. We have now included an additional experiment (Fig. S22B, also see left) which shows that the AAA lysine codon gives a much higher reporter activity than the AAG lysine codon, consistent with A-richness promoting efficient translation initiation. We had cited the Verma *et al.*, 2019 paper but this was accidentally removed in the submitted version, we apologise for this. We have now returned this citation and we have included the additional citations mentioned

above and other relevant citations to the manuscript.

3i. In sum, the paper presents a quite rich deep-seq data analysis, but the analyses feel premature and inappropriate to substantiate the conclusions.

We thank the reviewer for the constructive criticism, and we have made amendments based on this and other reviewers comments and we feel the paper is now much stronger.

REVIEWER COMMENTS

Reviewer #1 (Remarks to the Author):

This round of revisions effectively addressed my previous concerns, and I have no further comments.

Reviewer #2 (Remarks to the Author):

The manuscript improved from the original submission with both visual presentation (additional figures) as well as information (correct referencing and additional information). Authors should still avoid using commentary as "slightly stronger SD sequence" in explaining result as there is no real quantification of "slightly stronger" or similar descriptive comparisons.

Reviewer #3 (Remarks to the Author):

This is a revised manuscript, which I reviewed previously. The authors made efforts to respond to the criticism, but only explain without showing and including the data in the manuscript. In the following I will use the numeration of the authors from their point-by-point responses:

3b. Include table/plots in the supplementary information to show the 5'-P-site distance for each read length that has been used by calibration of the data.

3d. 100 most abundant genes have been used. This is an arbitrary number and the authors should show it actually for a much larger group of genes (see the analysis done in PMID 12270832; PMID: 26495981).

Along this line, were the peaks at the start considered in the calculation of the TE values? Strong peaks at the beginning/ends should be excluded (usually analysis are done with excluding 10-15 codons at 3' and 5' ends to avoid a bias). In particularly in bacteria this is a significant problem because of the overlapping genes and genes organized in operons. Thus, it is an important consideration (a must) in analyzing bacterial ribosome profiling data.

Overall the methods section is very laconic and does not reveal all this important details in the data analysis.

3f. I could not find the growth curves in the revised manuscript. The information of the OD if sample withdrawal is not sufficient. The growth curves should be included in the supplementary data.

Response to reviewer comments

Reviewer #1 (Remarks to the Author):

This round of revisions effectively addressed my previous concerns, and I have no further comments.

We thank reviewer 1 for the comment. It was much appreciated.

Reviewer #2 (Remarks to the Author):

The manuscript improved from the original submission with both visual presentation (additional figures) as well as information (correct referencing and additional information). Authors should still avoid using commentary as "slightly stronger SD sequence" in explaining result as there is no real quantification of "slightly stronger" or similar descriptive comparisons.

Thank you for bringing this to our attention, we have now modified the "slightly stronger SD sequence" (See line 196). We could not find any further examples of similar descriptive comparisons and so we have not made any other changes other than the above-mentioned example.

Reviewer #3 (Remarks to the Author):

This is a revised manuscript, which I reviewed previously. The authors made efforts to respond to the criticism, but only explain without showing and including the data in the manuscript. In the following I will use the numeration of the authors from their point-by-point responses:

3b. Include table/plots in the supplementary information to show the 5'-P-site distance for each read length that has been used by calibration of the data.

We had updated the supplementary information to show this in Figure S4. We have now also updated the manuscript to include a table showing the 5'-P-site distance for each read length that has been used by calibration of the data (Table 5). See table below:

Read length	23	24	25	26	27	28	29	30
Salmonella SL1344	8 (2)	8 (2)	9 (1)	10 (3)	11 (2)	12 (1)	13 (3)	14 (2)
Salmonella SJW1103	8 (2)	8 (2)	9 (1)	10 (3)	11 (2)	12 (1)	13 (3)	14 (2)
Listeria 10403S	11 (2)	11 (2)	12 (1)					

5' position in codons in brackets.

3d. 100 most abundant genes have been used. This is an arbitrary number and the authors should show it actually for a much larger group of genes (see the analysis done in PMID 12270832; PMID: 26495981).

We are not sure what is meant by this comment. If this is a reference to the top or bottom 100 genes by TE or mRNA abundance, we have stated the percentages in the text (line 324): 2.9% and 3.6% of *Salmonella* and *Listeria* translated genes, respectively. We didn't restrict our analysis to just

these groups but also performed global comparisons of expression and sequence features, which can be found in the extended data and supplementary figures (Lines 278-281, 325, 336, figures S12, S13, S18, S19).

Along this line, were the peaks at the start considered in the calculation of the TE values? Strong peaks at the beginning/ends should be excluded (usually analysis are done with excluding 10-15 codons at 3' and 5' ends to avoid a bias). In particularly in bacteria this is a significant problem because of the overlapping genes and genes organized in operons. Thus, it is an important consideration (a must) in analyzing bacterial ribosome profiling data.

We thank the reviewer for recognising the importance of dissecting overlapping coding regions in bacteria ribosome profiling data. This is precisely one of the major motivations for us to develop a ribosome profiling protocol that produces highly-phased data in bacteria, allowing one to accurately capture TE by only considering phased reads (i.e. genuine RPFs) in corresponding coding regions, thereby resolving the issue of overlapping coding regions, translated in different reading frame.

Specifically, we excluded out-of-frame reads from the RPF calculations and only in-frame reads were used in the calculations of protein synthesis of a specific ORF. We also did not include initiation and termination peaks. This was clarified in the materials and methods section (extended material, end of second paragraph). We ensured we analysed the data in this way to avoid bias in our data.

We apologize that we had not made it clearer in the previous reviewers' response.

Overall the methods section is very laconic and does not reveal all this important details in the data analysis.

We apologize that we have not included these details in the main text as in addition to adhere to word limit, we used previously published bioinformatic pipeline, which was cited in the main text. The requested bioinformatic details are included in the detailed extended material section.

3f. I could not find the growth curves in the revised manuscript. The information of the OD if sample withdrawal is not sufficient. The growth curves should be included in the supplementary data.

We apologies for not including the growth curve in the previous revision as this manuscript focuses on identifying novel regulatory elements for translation in bacteria rather than investigating translation at different growth stages. However, we appreciate that the reviewer's view on the importance of growth curves and have now included growth curves as for both *Salmonella* and *Listeria* in supplementary figure S1B.

We thoroughly agree with reviewer #2's comment. We apologize for the oversight of not correcting the statement in our previous submission. It is now amended (line 196).